# Language Semantic Graph Guided
# Data-Efficient Learning

**Wenxuan Ma**
Beijing Institute of Technology
wenxuanma@bit.edu.cn

**Shuang Li**✉
Beijing Institute of Technology
shuangli@bit.edu.cn

**Lincan Cai**
Beijing Institute of Technology
lincancai@bit.edu.cn

**Jingxuan Kang**
University of Liverpool
sgjkang3@liverpool.ac.uk

## Abstract

Developing generalizable models that can effectively learn from limited data and with minimal reliance on human supervision is a significant objective within the machine learning community, particularly in the era of deep neural networks. Therefore, to achieve data-efficient learning, researchers typically explore approaches that can leverage more related or unlabeled data without necessitating additional manual labeling efforts, such as Semi-Supervised Learning (SSL), Transfer Learning (TL), and Data Augmentation (DA). SSL leverages unlabeled data in the training process, while TL enables the transfer of expertise from related data distributions. DA broadens the dataset by synthesizing new data from existing examples. However, the significance of additional knowledge contained within labels has been largely overlooked in research. In this paper, we propose a novel perspective on data efficiency that involves exploiting the semantic information contained in the labels of the available data. Specifically, we introduce a **Language Semantic Graph** (**LSG**) which is constructed from labels manifest as natural language descriptions. Upon this graph, an auxiliary graph neural network is trained to extract high-level semantic relations and then used to guide the training of the primary model, enabling more adequate utilization of label knowledge. Across image, video, and audio modalities, we utilize the LSG method in both TL and SSL scenarios and illustrate its versatility in significantly enhancing performance compared to other data-efficient learning approaches. Additionally, our in-depth analysis shows that the LSG method also expedites the training process.

## 1 Introduction

Deep learning has achieved remarkable breakthroughs in various domains, including speech recognition, visual object recognition, object detection, drug discovery, genomics, and many others [33]. However, the increasing size of deep neural networks has led to a growing demand for large volumes of data. Unfortunately, not all application domains have access to extensive datasets due to the high costs associated with data collection and human annotation [38, 8, 16]. Consequently, substantial efforts have been dedicated to studying methods that enable data-efficient learning [1, 49], aiming to alleviate the requirement of extensive training data and, in particular, human supervision.

The current primary approaches to achieving data-efficient learning encompass semi-supervised learning (SSL) [62], transfer learning (TL) [49, 86], and other techniques such as data augmentation

---

✉ Corresponding author. Code available at: `https://github.com/BIT-DA/LSG`

(DA) [78, 9]. Semi-supervised learning [19, 34, 58, 4, 69] involves using both labeled and unlabeled data to reduce manual labeling effort and enhance model performance. By exploring vast amounts of unlabeled samples, artificial learners acquire a more comprehensive understanding of the underlying data structure and achieve improved predictions, even in scenarios where labeled samples are scarce [70]. Transfer learning leverages data from other related source domains typically offer abundant data and annotations [42, 14, 2], or it utilizes models pretrained on large-scale datasets to transfer general knowledge to minimize the data requirements in the target domain [76, 74]. Data augmentation technique creates novel examples from existing ones, thereby expanding and diversifying the training dataset. From this perspective, these data-efficient learning approaches share a common objective: to leverage additional training data without necessitating extra manual labeling efforts. In this paper, we approach data-efficient learning from a different angle: by investigating improved methods to harness the available data-label pairs. This novel perspective motivates the core objective of our work: *to fully exploit semantic information contained in the labels to improve model performance with data efficiency.*

In particular, it is worth noting that original human annotations typically manifest as natural language descriptions of the training data, such as "bicycle" and "kangaroo" in object recognition tasks or "jogging" and "playing piano" in action recognition. We refer to these descriptive labels with descriptions as the labeled concepts. The prevailing approach of converting these labeled concepts into indices simplifies the machine learning problem but sacrifices semantic information. Consequently, our objective is to enhance data utilization by reintegrating this information into the training process. Our idea involves introducing an auxiliary branch during training to facilitate high-level interactions between data representations and label semantics. This interaction guides the model in extracting appropriate features that capture accurate semantics. Notably, the auxiliary branch is discarded after the training process, thus avoiding an increase in inference cost. However, the question that remains is how to achieve such interaction effectively.

To this end, we propose employing graphs to model high-level semantics of the labeled concepts, owing to their generality and flexibility [72]. The initial node features within the graph comprise language embeddings of these concepts, which are generated by pretrained language models. The edges of the graph capture strong semantic connections between the concepts. We refer to this graph as the Language Semantic Graph (LSG) and train an auxiliary graph neural network to extract the knowledge within the graph and transfer it to the primary neural network model. During formal training, the data representations produced by the primary model are incorporated into the Language Semantic Graph as new nodes, based on their true labels. Consequently, the graph neural network can propagate information across all nodes in the augmented graph. Building upon this interaction mechanism, we propose two additional optimization objectives to implicitly align and explicitly regularize the data representations, ensuring their adherence to the correct semantics.

LSG demonstrates applicability across models in various modalities and exhibits improved data-efficient learning performance. To validate its effectiveness, we conduct experiments on two typical data-efficient scenarios: transfer learning and semi-supervised learning, and apply it to image, video, and audio datasets. The experimental results demonstrate that LSG significantly enhances performance and outperforms other data-efficient approaches. Our contribution includes:

- We propose a novel perspective towards data-efficient learning via enhancing the utilization efficiency on the available annotated data. Particularly, we introduce a Language Semantic Graph (LSG) to model the high-level label semantics and provide extra supervision.
- We design a comprehensive method with an auxiliary graph neural network that extracts knowledge from LSG and two novel objectives that guide the primary model to learn efficiently. Furthermore, we extend the method by incorporating semi-supervised learning to leverage unlabeled samples for further enhancement.
- Experiments on both transfer learning and semi-supervised learning scenarios are conducted, crossing image, video and audio modalities. The superior performance achieved by our proposed method, along with an in-depth analysis, confirms its effectiveness.

## 2 Related Work

**Data-efficient learning** [1] investigates the problem of data hungriness of modern machine learning algorithms, and proposes algorithms to reduce the reliance on extensive training data. Two

prominent approaches in the roadmap of data-efficient learning are Transfer Learning (TL) and Semi-Supervised Learning (SSL). TL focuses on transferring knowledge from source domains to minimize the number of labeled examples required in a target domain [86]. Within this research area, Domain Adaptation [14, 55, 82, 35] investigates transductive transfer learning where source and target domain possess different data distributions but share the same task. Fine-tuning have emerged with deep neural networks [12, 48, 76, 18], enabling the flexible transfer of knowledge from large pretrained models to benefit various downstream tasks [56, 74, 41, 23, 61]. In conclusion, TL generally leverages data from elsewhere to enhance data efficiency in the current task.

On the other hand, SSL takes a different approach by harnessing large-scale unlabeled data to reduce the need for manual labeling. Prevalent SSL methods [34, 58, 60, 3, 24] explore unlabeled samples to capture the intrinsic data structural information [70]. Notably, a widely adopted strategy in SSL is the assignment of pseudo-labels to samples [34], enabling models to improve in a self-training manner [53, 5, 75]. The combination of TL and SSL approaches has also been embraced to achieve more promising results [54, 67]. In contrast, our work presents a novel perspective within data-efficient learning. Instead of exploring data from various sources, we aim to exploit annotated data more comprehensively. We identify semantic relations between labels and utilize them to guide the model in learning better representations.

**Graph neural network as a learning component**. Graph neural networks (GNN) [28, 17, 63, 64] have gained significant popularity not only for their standard applications in graph-related tasks such as node classification and link prediction but also as a component within learning systems to process graph-structured inputs and propagate crucial information [15, 71, 50, 45]. In the domain of computer vision, Garcia and Bruna [15] leverage GNN for few-shot learning, NGM [20] generates graph with continuously updated adjacency matrix for 3D action recognition, BGRNet [71] includes GNN in their framework for panoptic segmentation, and Knowledge-CLIP [50] learns multi-modal representations. Furthermore, GNNs have found applications in natural language processing [45], recommendation systems [46] and etc. In our method, a GNN is trained on the semantic graph constructed from labels to guide the learning process of the primary model.

**Language models for non-language tasks**. The beneficial interaction between modalities, justified by the success of multimodal learning [51, 79, 85, 57], has sparked growing interest in borrowing knowledge from other modalities. LIFT [11] formalizes non-language inputs into sentences and adopts GPT [52] as general interface. Vision-Language model-based finetuning methods [84, 83, 43, 81, 80] leverage the jointly pretrained text encoders to construct robust classifiers for downstream vision tasks. Our method draws inspiration from these works but differs in its objective as well as the way to leverage linguistic knowledge. Concretely, LSG promotes data-efficient learning on independent non-language models that are not restricted to multi-modal model. Furthermore, instead of building a robust classification head, the linguistic knowledge in our method is leveraged to construct a graph that models label semantic relations then guide the non-language model learn more semantically meaningful representations .

## 3   Method

Training a deep neural network to achieve satisfying performance could be challenging in many real-world applications due to the lack of sufficient label supervision. Our approach seek to improve the utilization of labeled data that is currently available for enhancing data efficiency. Formally, we consider training a model on a task-related labeled dataset $\{\boldsymbol{x}_i, y_i\}_{i=1}^{n}$, where $\boldsymbol{x}$ could be images, video clips or audio clips etc. for different tasks and $y$ is the corresponding category label that is represented using natural language description such as a word or a phrase. The model (also referred as the **primary model** in the following content) can be regard as a feature encoder $F(\cdot|\theta)$ that maps the input data to embedding space and follows by a task-specific head $C(\cdot|\phi)$ that turns feature embeddings into predictions, with $\theta$ and $\phi$ as parameters. The standard training objective utilizing the labeled dataset is the minimization of the empirical risk. By omitting the network parameters and denote $C \circ F(\boldsymbol{x}) = C(F(\boldsymbol{x}))$, the empirical risk minimization can be formulate as:

$$\min_{\theta,\phi} \quad \mathcal{L}_{emp} = \frac{1}{n} \sum_{i=1}^{n} \ell(C \circ F(\boldsymbol{x}_i), y_i), \tag{1}$$

where $\ell(\cdot, \cdot)$ typically be cross-entropy loss, and labels are converted to indices during computation.

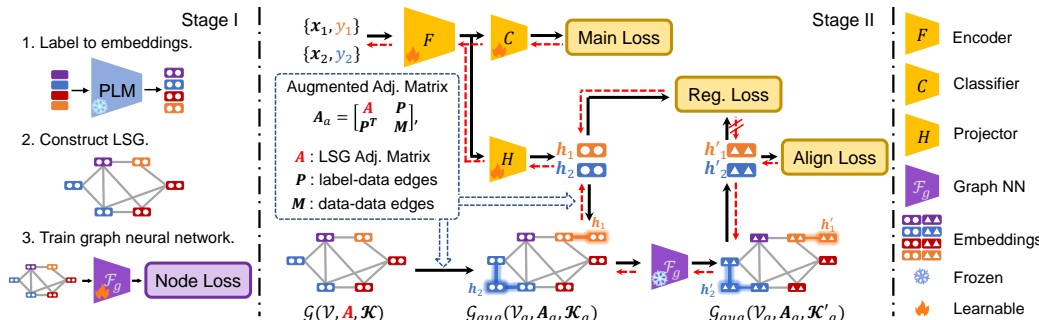

Figure 1: Illustration of the LSG training framework. In the first stage, labeled concepts are turned into text embeddings via pretrained language model (PLM), and then the Language Semantic Graph (LSG) is constructed according to the embeddings. We train a GNN on the graph to extract semantic knowledge. In the second or the formal training stage, the data representations $\boldsymbol{h}$ are connected to the graph based on their labels (and the new graph topology is described in the augmented adjacency matrix). Representations are then transformed into $\boldsymbol{h}'$ by the GNN. The proposed alignment loss and regularization loss is applied on these features to guide the primary model.

We argue that such conversion of labels inevitably losses information and results in data inefficiency. Therefore, our method introduces two additional losses $\mathcal{L}_{align}$ and $\mathcal{L}_r$ that exploit the discarded label semantic information for complement.

This section first introduces the construction process of the Language Semantic Graph (LSG) and the training process of the auxiliary model, which are conducted before the primary training stage to capture the semantic information. Then it moves on to explain the objectives $\mathcal{L}_{align}$ and $\mathcal{L}_r$ guide the data-efficient training of the primary model through LSG.

## 3.1 Language Semantic Graph

LSG aims to model the semantic relationship between labeled concepts with high generality and flexibility. We propose a straightforward approach to construct the graph, leveraging the text embedding space that contains rich semantics. Specifically, given the natural language descriptions of the concepts $\{\mathcal{W}_k\}_{k=1}^K$ where $K$ is the total number of concepts or categories, we use a set of $m$ predetermined prompts (*e.g.*, 'This is a photo of a {concept}') to contextualize the concepts. We then obtain a set of text embeddings $T$ by inputting contextualized concepts into a frozen pretrained language model such as BERT [25], where $T = \cup_{k=1}^K \{\boldsymbol{t}_k^{(1)}, \boldsymbol{t}_k^{(2)}, ..., \boldsymbol{t}_k^{(m)}\} \subseteq \mathbb{R}^{d_t}$, $|T| = mK$, with $d_t$ being the dimension of the text embedding space. Then we define the text embedding matrices $\mathcal{K}$ and $\tilde{\mathcal{K}}$, such that $\mathcal{K} = [\boldsymbol{t}_1^{(1)}, ..., \boldsymbol{t}_1^{(m)}, \boldsymbol{t}_2^{(1)}, ..., \boldsymbol{t}_K^{(m)}] \in \mathbb{R}^{d_t \times |T|}$ and $\tilde{\mathcal{K}}$ be its column normalized matrix. We compute the cosine similarity matrix between text embeddings as $\boldsymbol{S} = \tilde{\mathcal{K}}^\top \tilde{\mathcal{K}}$. Additionally, we define a text label vector $\boldsymbol{y}_t = [\underbrace{1, ..., 1}_{m}, ..., \underbrace{K, ..., K}_{m}]$ to indicate the category for each text embedding.

Next, we define the language semantic graph adjacency matrix based on text embedding similarities:

$$\boldsymbol{A} = [a_{ij}]_{1 \leq i \leq |T|, 1 \leq j \leq |T|} = \begin{cases} \max(\boldsymbol{S}_{i,j}, 0) & \text{if } y_{t,i} = y_{t,j}, \\ \boldsymbol{S}_{i,j} & \text{if } y_{t,i} \neq y_{t,j} \text{ and } \boldsymbol{S}_{i,j} \geq \tau, \\ 0 & \text{otherwise.} \end{cases} \quad (2)$$

Note that $\boldsymbol{A}$ is a weighted adjacency matrix. The first line means that we connect nodes representing the same concept with their similarity scores as weight. The second line allows certain edge connections between nodes that represent different concepts but share high feature similarities, which is controlled be an edge threshold hyper-parameter $\tau$. Finally, the language semantic graph we defined can be represented as: $\mathcal{G} = (\mathcal{V}, \boldsymbol{A}, \mathcal{K})$, where $\mathcal{V}$ is the node set, $|\mathcal{V}| = |T|$, and $\mathcal{K}$ is used as the node feature matrix.

To fully leverage the LSG to guide the data-efficient training of the primary model, we propose to train an **auxiliary graph neural network** to extract semantic information from LSG. We adopt the

standard GCN [28] architecture, which consists of several consecutive graph convolution layers. It is expected that GCN, with its strength in massage passing, first refines the initial node embeddings (or text embeddings) by leveraging graph topology and then serves as a bridge connecting the primary model and the LSG. Let $D$ be the degree matrix of $A$, $\sigma(\cdot)$ be the ReLU [47] activation function, $W^{(l)}$ be the learnable weight matrix for the $l_{\text{th}}$ layer and $\mathcal{K}^{(0)} = \mathcal{K}$, the update function of the node features can be formulated as:

$$\mathcal{K}^{(l)} = \sigma(D^{-\frac{1}{2}} A D^{-\frac{1}{2}} \mathcal{K}^{(l-1)} W^{(l)}). \tag{3}$$

Analogues to the primary model, we regard our GCN model as the composition of node feature encoder $\mathcal{F}_g$ and node classifier $\mathcal{C}_g$. Particularly, the encoder $\mathcal{F}_g$ is designed to output node features that have the same dimension as input features for the later computation of $\mathcal{L}_r$. The whole network is trained using the node classification objective that aims to correctly predict category for each node:

$$\mathcal{L}_{node} = -\sum_{i=1}^{|T|} \sum_{k=1}^{K} \mathbb{1}_{[k=y_{t,i}]} \log \delta\left(\mathcal{C}_g \circ \mathcal{F}_g(\mathcal{K}_{\cdot,i})\right)_k, \tag{4}$$

where $\delta(\cdot)$ is the softmax function. To this end, we obtain the LSG and the auxiliary model trained on it, which contain semantic information from the labeled concepts and will be leveraged to boost the data-efficient training of the primary model.

**Remark**: $\mathcal{L}_{node}$ is only used for training GCN in this first stage. By doing this, each node processed by the GCN aggregates discriminative and more comprehensive semantic information from neighboring nodes. Also note that both the LSG and the GCN only depend on the categories of the current task. Hence, they can be constructed and trained on each task once and for all.

### 3.2 LSG Guided Data-Efficient Learning

To enable language semantic guidance, we add a new projector $H(\cdot|\psi)$ on top of the feature encoder $F$ of the primary model to project the data representation to the same dimension as the text embeddings. We denote $\boldsymbol{h}_i = H(F(\boldsymbol{x}_i)) \in \mathbb{R}^{d_t}$ as the projected representation of $\boldsymbol{x}_i$ and $\tilde{\boldsymbol{h}}$ as its embedding after $l_2$ normalization. We would like to enable interactions between the data representations and the node features within the semantic graph. Henceforth, we propose to connect features generated by the primary model into LSG according to the label $y$ of the data.

During training process, for every minibatch of $B$ images $\{\boldsymbol{x}_j, y_j\}_{j=1}^{B}$, we obtain their representations $\{\boldsymbol{h}_i\}_{i=1}^{B}$ and construct an augmented graph of the original Language Semantic Graph, denoted as $\mathcal{G}_{aug} = (\mathcal{V}_a, \boldsymbol{A}_a, \mathcal{K}_a)$. In this augmented graph, $|\mathcal{V}_a| = |T| + B$, the augmented node feature matrix $\mathcal{K}_a = [\mathcal{K}, \boldsymbol{h}_1, ..., \boldsymbol{h}_B] \in \mathbb{R}^{d_t \times (|T|+B)}$ is the node feature matrix plus $B$ data features, and the augmented adjacency matrix is defined as $\boldsymbol{A}_a = \begin{bmatrix} \boldsymbol{A} & \boldsymbol{P} \\ \boldsymbol{P}^\top & \boldsymbol{M} \end{bmatrix}$, where $\boldsymbol{P} \in \mathbb{R}^{|T| \times B}$ is a binary matrix indicating the connection between label nodes and data nodes, $\boldsymbol{P} = [p_{ij}] = \begin{cases} 1 & \text{if } y_{t,i} = y_j, \\ 0 & \text{otherwise,} \end{cases}$ and $\boldsymbol{M} \in \mathbb{R}^{B \times B}$ is a symmetric binary matrix indicating the connection among data nodes, $\boldsymbol{M} = [m_{ij}] = \begin{cases} 1 & \text{if } y_i = y_j, \\ 0 & \text{otherwise.} \end{cases}$ Based on the true label of training data, these edges build connection between the primary model and the LSG, which allows the graph neural network to propagate information and transfer knowledge that is embedded inside the labels.

With the augmented graph, we conduct language semantic guidance to the primary model through optimizing two objectives: $\mathcal{L}_{align}$ that implicitly aligns the data representation to the semantic graph and $\mathcal{L}_r$ that explicitly regularize the model to extract more semantically meaningful representations. We utilize the frozen auxiliary model to encode new data nodes in the augmented graph and predict their categories. Given that the trained auxiliary model can correctly predict the label nodes in the original graph and its weights are no longer updated, the data representations $\boldsymbol{h}_i$ are forced to become similar to their corresponding label node features during optimization, achieving the alignment between features from different node. The implicit alignment objective is as follows:

$$\min_{\theta, \psi} \quad \mathcal{L}_{align} = -\sum_{i=1}^{B} \sum_{k=1}^{K} \mathbb{1}_{[k=y_i]} \log \delta(\mathcal{C}_g \circ \mathcal{F}_g(\boldsymbol{h_i}))_k. \tag{5}$$

Besides the implicit alignment loss that guides the data representations to have correct semantic relations, we further propose an explicit regularization loss to transfer the knowledge embedded

in the label nodes to the primary model. Notice that after being processed by the trained GCN, the output features of the data nodes have already aggregated information from their label node neighbors on the augmented graph, which are desirable representations that contain information from both the data samples and the semantic graph. Thus, we propose the graph semantic feature regularization loss $\mathcal{L}_r$ that directly uses the output features of the auxiliary model encoder $\mathcal{F}_g$ to promote their initial input features, which is why both features are designed to have the same dimension. Specifically, we denote $\mathcal{F}_g(\boldsymbol{h_i})$ as the output graph feature for $\boldsymbol{h_i}$. Then we normalize both features $\mathcal{F}(\tilde{\boldsymbol{h_i}}) = \mathcal{F}_g(\boldsymbol{h_i})/||\mathcal{F}_g(\boldsymbol{h_i})||_2$, $\tilde{\boldsymbol{h}}_i = \boldsymbol{h_i}/||\boldsymbol{h_i}||_2$ and use the normalized graph feature as the target for normalized input feature. We have the following regularization objective:

$$\min_{\theta, \psi} \quad \mathcal{L}_r = -\sum_{i=1}^{B} ||\text{sg}(\mathcal{F}_g(\tilde{\boldsymbol{h_i}})) - \tilde{\boldsymbol{h_i}}||_2^2, \tag{6}$$

where $\text{sg}(\cdot)$ denotes stop gradient. By explicitly regularizing the original data representation to include more information embedded in the graph as Eq. (6), the primary model learns more informative features from the LSG and thus allows this auxiliary branch to be discarded during inference.

Two proposed objectives $\mathcal{L}_{align}$ and $\mathcal{L}_r$ work in a complementary manner to convey the semantic relations between labeled concepts to primary model, whereas the classical empirical risk minimization helps the model to discriminate. Combining the three losses, our proposed method achieves fully exploitation of the information inside human annotations, and improves the data utilization efficiency. The final optimization objective of LSG is the weighted sum of the three losses joined by trade-off parameters $\lambda$ and $\mu$. The full two-stage training process is shown in Fig. 1.

$$\min_{\theta, \phi, \psi} \quad \mathcal{L} = \mathcal{L}_{emp} + \lambda\mathcal{L}_{align} + \mu\mathcal{L}_r. \tag{7}$$

**Remark**: The GCN and correspondingly the projector is not used during inference, and the classification result is still produced by the primary model. For this reason, no extra computational cost is needed in inference time.

### 3.3  Extend LSG to Unlabeled Data

We further discuss a straightforward extension of the proposed LSG on unlabeled training dataset $\{\boldsymbol{u}\}$ that share a same label space with the labeled data . By simply assigning a hard pseudo-label to each of them, these unlabeled data can also be connected to the semantic graph and be leveraged to guide the model. We find surprisingly that the vanilla Pseuso-Labeling strategy [34] works satisfactory with our method in semi-supervised learning. To be specific, we assign pseudo-labels to all unlabeled data according to the maximum class in model output: $\hat{y} = \arg\max_c \delta(C \circ F(\boldsymbol{u}))$, and these pseudo-labels are updated after each epoch. To this end, we extend LSG to semi-supervised learning by jointly utilize labeled and unlabeled data for constructing the augmented semantic graph, and the objective now becomes

$$\min_{\theta, \phi, \psi} \quad \mathcal{L}_{ssl} = \mathcal{L}_{emp}(\boldsymbol{x}) + \lambda\mathcal{L}_{align}(\boldsymbol{x}, \boldsymbol{u}) + \mu\mathcal{L}_r(\boldsymbol{x}, \boldsymbol{u}). \tag{8}$$

With these pseudo-labeled samples, our method achieves significant performance boost despite that there exists noisy pseudo-labels. We hypothesis that it is because the pseudo-labels are leveraged only for the auxiliary branch and hence do not bias the classification head, as also suggested in [5]. We believe that further improvements can be achieved via combining our method with more advanced pseudo-labeling strategies in SSL, which will be left as future works.

## 4  Experiments

**Datasets and Models**. We conduct experiments on 7 standard datasets that are intensively studied in Transfer Learning [77, 41, 27] and Semi-supervised learning [67, 24] and cover input data ranging images, videos and audios. For image datasets, we adopt *FGVC Aircraft* [44] (10,000 images for 100 aircraft variants), *Stanford Cars* [30] (16,185 images for 196 car categories) and *CUB-200-2011* [66] (11,788 images for 200 bird species) for fine-grained classification analysis and *Office Home* [65] (four domains, each contains roughly 4,000 images for 65 categories) to evaluate out-of-distribution performance. For video datasets, we use *UCF-101* [59] (13,320 video clips in 101 categories) and *HMDB51* [31] (6,766 clips form 51 actions) For audio dataset, we report the performance on *AudioSet-20K* [16].

Table 1: Classification accuracy (%) on three fine-grained image benchmarks under fully-supervised and semi-supervised learning settings. ResNet-50 pretrained on ImageNet-1k is adopted as backbone and SSL indicates whether the rest training data is leveraged as unlabeled samples.

| Method | SSL | FGCV Aircraft | | | | Stanford Cars | | | | CUB-200 | | | |
|---|---|---|---|---|---|---|---|---|---|---|---|---|---|
| | | 15% | 30% | 50% | 100% | 15% | 30% | 50% | 100% | 15% | 30% | 50% | 100% |
| Fine-tuning | ✘ | 41.6 | 57.8 | 68.7 | 80.2 | 41.1 | 65.9 | 78.4 | 87.8 | 51.2 | 64.6 | 74.6 | 81.8 |
| LWF [37] | ✘ | 44.1 | 60.6 | 68.7 | 82.4 | 44.9 | 67.0 | 77.6 | 87.5 | 56.7 | 66.8 | 73.4 | 81.5 |
| DELTA [36] | ✘ | 43.6 | 59.5 | 69.6 | 81.2 | 43.3 | 67.6 | 79.6 | 88.0 | 53.4 | 66.7 | 76.0 | 82.0 |
| BSS [7] | ✘ | 44.4 | 61.9 | 71.4 | 82.7 | 45.0 | 68.4 | 79.6 | 88.4 | 54.8 | 67.3 | 76.3 | 82.3 |
| StochNorm [29] | ✘ | 44.3 | 60.6 | 70.1 | 81.5 | 44.4 | 68.1 | 79.3 | 87.9 | 54.8 | 66.8 | 75.8 | 82.2 |
| Co-Tuning [77] | ✘ | 45.9 | 61.6 | 72.7 | 83.9 | 49.0 | 70.6 | 81.9 | 89.5 | 57.6 | 70.1 | 77.3 | **82.5** |
| **LSG** | ✘ | **55.6** | **72.0** | **79.5** | **86.7** | **55.4** | **75.5** | **83.8** | **90.7** | **57.7** | **70.6** | **77.5** | 82.2 |
| Pseudo-Labeling [34] | ✔ | 46.8 | 62.8 | 73.2 | – | 40.9 | 67.0 | 78.7 | – | 45.3 | 62.0 | 72.3 | – |
| FixMatch [58] | ✔ | 55.5 | 71.6 | 78.3 | – | 49.9 | 77.5 | 84.8 | – | 44.1 | 63.5 | 76.0 | – |
| SimCLRv2 [6] | ✔ | 40.8 | 59.0 | 68.5 | – | 45.7 | 61.7 | 77.5 | – | 45.7 | 62.7 | 71.1 | – |
| Self-Tuning [67] | ✔ | 64.1 | 76.0 | 81.2 | – | 72.5 | 83.6 | 88.1 | – | 64.2 | 75.1 | 80.2 | – |
| DebiasMatch [68] | ✔ | 59.5 | 71.2 | 77.1 | – | 75.3 | 86.1 | 90.0 | – | 64.7 | 75.1 | 77.7 | – |
| **LSG (extended)** | ✔ | **71.4** | **85.3** | **87.1** | **88.6** | **79.3** | **87.7** | **90.9** | **91.9** | **66.1** | **76.1** | **80.6** | **82.6** |

Table 2: Classification accuracy (%) with self-supervised pretraining model (CLIP ViT-B).

| Method | FGCV Aircraft | | | | Stanford Cars | | | | CUB-200 | | | |
|---|---|---|---|---|---|---|---|---|---|---|---|---|
| | 15% | 30% | 50% | 100% | 15% | 30% | 50% | 100% | 15% | 30% | 50% | 100% |
| CoOp [84] | 34.8 | 38.2 | 42.9 | 49.8 | 75.0 | 81.3 | 83.7 | 84.0 | 64.4 | 72.3 | 76.3 | 78.7 |
| ProDA [43] | 33.9 | 36.6 | 43.5 | 50.8 | 75.4 | 81.5 | 84.0 | 84.2 | 65.5 | 73.1 | 76.4 | 79.2 |
| Tip-Adapter [81] | 36.9 | 41.7 | 46.7 | 53.8 | 75.6 | 81.4 | 84.1 | 85.0 | 69.0 | 74.9 | 77.9 | 81.2 |
| **LSG** | **48.9** | **58.6** | **65.0** | **74.5** | **79.4** | **83.2** | **86.1** | **90.1** | **70.2** | **78.4** | **81.9** | **85.4** |

Several deep neural networks with different architecture and pretraining dataset are included in the experiments. For image tasks, we use ResNet-50 [21] and ConvNext-S [40] pretrained on ImageNet-1k [10], ViT-B pretrained on CLIP WIT [51], and Swin-B [39] pretrained on ImageNet-21k. For video and audio tasks, we adopt pretrained ViT-L [13] from VideoMAE [61] and ViT-B from Audio-MAE [22], respectively.

**Implementation Details**. We utilize a pretrained BERT-L [25] as the language model to transform labels to text embeddings. For each concept, we contextualize it into complete sentences using 20 handcrafted prompts. When constructing the Language Semantic Graph, the similarity threshold $\tau$ is determined adaptively to include the top $\rho = 0.3\%$ edges that connects between nodes of different labels of the fully connected graph. The GCN is trained on full graph for 5,000 iterations.

For LSG guided training process, the projector $H$ is implemented by a fully-connected layer with an output dimension of 1024 and randomly initialized weights. We find that setting $\lambda$ and $\mu$ to 1.0 and 8.0 generally achieves satisfying results within all the experiments. In image classification tasks, we adopt SGD with a momentum of 0.9 as the optimizer. The learning rate is set as 1e-3 for the visual backbone in most experiments and a $10\times$ larger value is applied for the classifier and projector in SSL and SDG. In video and audio tasks, we adopt the same configurations as finetuning in the official released VideoMAE and Audio-MAE codebase. Please refer to the appendix for more details.

## 4.1 Results of Image Experiments

To examine the effectiveness of our method, we first conduct experiments on three fine-grained image classification benchmarks *FGVC-Aircraft*, *Stanford Cars* and *CUB-200-2011* following [77, 67]. Similar to previous works, we analyze the performance of LSG under labeled data partition ratio of 15%, 30%, 50% as well as the full training set. We adopt ImageNet-1k pretrained ResNet-50 as backbone and compare LSG with previous transfer learning baselines: vanilla fine-tuning, LWF [37], DELTA [36], BSS [7], StochNorm [29] and Co-Tuning [77]. In addition, to show that LSG is applicable to self-supervised pretraining models, we adopt CLIP pretrained ViT-B and compare LSG to vision-language fine-tuning methods CoOP [84], ProDA [43], Tip-Adapter [81]. Finally, we extend LSG to semi-supervised learning setting by including the rest training data as unlabeled data. As discussed in § 3.3, we adopt Pseudo-Labeling [34] strategy for unlabeled data (which can be regarded as the baseline for extended LSG) and compare the extended LSG with SSL methods including FixMatch [58], DebiasMatch [68], SimCLRv2 [6] and Self-Tuning [67].

Table 3: Out-of-distribution accuracy (%) for single domain generalization on *Office-Home*. Backbone ConvNext-S and Swin-B are pre-trained on ImageNet-1k and ImageNet-22k respectively.

| Backbone | Method | Source:Ar | | | | Source:Cl | | | | Source:Pr | | | | Source:Rw | | | | Avg. (ID) | Avg. (OOD) |
|---|---|---|---|---|---|---|---|---|---|---|---|---|---|---|---|---|---|---|---|
| | | Ar | Cl | Pr | Rw | Cl | Ar | Pr | Rw | Pr | Ar | Cl | Rw | Rw | Ar | Cl | Pr | | |
| ConvNext-S | ERM [27] | 85.0 | 53.4 | 72.7 | 78.6 | 85.0 | 67.5 | 72.9 | 75.4 | 94.7 | 61.8 | 49.0 | 80.0 | 91.4 | 72.2 | 52.7 | 80.9 | 89.0 | 68.1 |
| | LP-FT [32] | 85.1 | 55.3 | 73.3 | 79.5 | 85.6 | 70.4 | 73.4 | 77.5 | 95.1 | 63.9 | 52.0 | 82.0 | 91.4 | 72.0 | 54.1 | 82.2 | 89.3 | 69.6 |
| | LSG | 85.8 | 57.7 | 74.0 | 79.9 | 86.3 | 71.7 | 75.4 | 77.8 | 95.1 | 65.8 | 54.7 | 82.1 | 91.2 | 73.8 | 58.0 | 82.3 | 89.6 | 71.1 |
| Swin-B | ERM [27] | 91.8 | 70.7 | 86.1 | 88.5 | 89.1 | 80.6 | 84.3 | 86.7 | 96.5 | 77.9 | 66.1 | 88.3 | 95.2 | 82.6 | 69.1 | 90.4 | 93.2 | 81.0 |
| | LP-FT [32] | 91.5 | 70.4 | 85.9 | 88.6 | 89.4 | 81.3 | 85.4 | 87.1 | 96.7 | 78.8 | 68.0 | 88.9 | 95.1 | 82.9 | 70.3 | 90.5 | 93.2 | 81.5 |
| | LSG | 92.4 | 72.4 | 85.7 | 88.3 | 89.8 | 83.7 | 87.1 | 88.3 | 96.7 | 80.9 | 69.9 | 89.4 | 95.1 | 83.0 | 70.8 | 90.5 | 93.5 | 82.5 |

The results are shown in Table 1 and Table 2. We observe that our method outperforms its counterparts on each task and in both fully-supervised and semi-supervised scenarios. In fully supervised experiments, LSG significant outperforms other methods on *FGVC-Aircraft* and *Stanford Cars* especially when the available labeled data is more scarce. It also achieves comparable performance on *CUB200*. At 15% sampling ratio, LSG improves from the vanilla fine-tuning by 14.0%, 14.3% and 6.5% on three datasets, and surpasses Co-tuning by an average accuracy of 5.3%. Similarly, LSG shows promising potential in semi-supervised setting, achieving the best performance across all of the labeling rate and all three datasets. Notably, since the extended LSG incorporates data augmentation techniques adopted in semi-supervised learning, its results improvement over LSG under 100% training data shows that the proposed method is mutually beneficial to other data-efficient strategies. When applied on self-supervised pretrained model, LSG also demonstrates consistent gains comparing to fine-tuning methods that are specifically designed for vision-language models.

We also conduct experiments on cross-domain benchmark *Office-Home* to evaluate the out-of-distribution performance of the LSG guided training. Following [27], we consider the challenging single domain generalization setting where models are trained on one single source domain and tested on the rest target domains without seeing any data from them. We compare LSG against the vanilla empirical risk minimization (ERM) baseline and the more advanced approach LP-FT [32].

We report the prediction accuracies of both in-distribution (ID) and out-of-distribution (OOD) in Table 3. Specifically, samples in the source domain are randomly partitioned into 80% training data and 20% test data, and the results refers to accuracy on test data. OOD results are obtained evaluating all data in each target domain. The results show that LSG improves model performance on both ID and OOD samples, demonstrating that the label semantic relations help the model learn features that are more robust to distribution shift.

## 4.2 Results of Video and Audio Experiments

Category labels in video and audio tasks also have their natural language descriptions, such as "push up", "kiss" in *HMDB51* dataset and "singing", "groan" in *AudioSet*. This enables LSG to construct semantic graph and improve the model training process. We adopt VideoMAE pretrained ViT-L as backbone for small-scaled video datasets *UCF-101* and *HMDB51*, and use Audio-MAE pretrained ViT-B for audio benchmark *AudioSet*.

Table 4: Accuracies (%) of video action recognition on *UCF-101* and *HMBD51* under label rate of 1% and 10%. Backbone: ViT-L pretrained by VideoMAE.

| Method | SSL | UCF-101 | | HMDB51 | |
|---|---|---|---|---|---|
| | | 1% | 10% | 1% | 10% |
| VideoMAE ft [61] | ✗ | 26.80 | 73.26 | 11.82 | 46.85 |
| LSG | ✗ | 29.07 | 74.49 | 13.28 | 50.50 |
| Pseudo-Labeling [34] | ✔ | 28.11 | 76.08 | 11.97 | 47.72 |
| FixMatch [58] | ✔ | 30.51 | 76.45 | 13.12 | 48.30 |
| CMPL [73] | ✔ | 32.24 | 77.82 | 13.58 | 50.81 |
| LSG (extended) | ✔ | 35.32 | 80.36 | 15.40 | 52.65 |

Following previous data efficient studies [24, 73] on the two video benchmarks, we train the model using 1% and 10% labeled data respectively. We also conduct SSL experiments similar to image experiments and compare LSG against SSL approaches in video. From the results in Table 4, we observe that LSG consistently improves the fine-tuning accuracy over

Table 5: Classification accuracies (%) on *AudioSet-20K* with label proportion of 10%, 25% and 50%. (ft: Fine-tuning)

| Method | 10% | 25% | 50% |
|---|---|---|---|
| Audio-MAE ft [22] | 2.59 | 15.60 | 25.96 |
| LSG | 11.20 | 21.80 | 27.82 |

all tasks with only limited amount of labeled samples available. With unlabeled data being included, two classical SSL methods Pseudo-Labeling and FixMatch both achieve certain improvements, whereas CMPL [73], a method designed specifically for video SSL tasks, performs better. On top of that, LSG obtains superior results than these method on all of the tasks, boosting the accuracy from

Table 6: Ablation study of losses and alternative design choices on *FGVC Aircraft*, 15% labeled data.

| Method | $\mathcal{L}_{emp}$ | $\mathcal{L}_{align}$ | $\mathcal{L}_r$ | $\mathcal{L}_r$ w/ sg. | classifier | language head | Acc. |
|---|---|---|---|---|---|---|---|
| LSG w/o $\mathcal{L}_r$, $\mathcal{L}_{align}$ | ✔ | | | n/a | ✔ | | 41.6 (-14.0) |
| LSG w/o $\mathcal{L}_r$ | ✔ | ✔ | | n/a | ✔ | | 44.2 (-11.4) |
| LSG w/o $\mathcal{L}_{align}$ | ✔ | | ✔ | ✔ | ✔ | | 50.7 (-4.9) |
| LSG | ✔ | ✔ | ✔ | ✔ | ✔ | | **55.6** (+0.0) |
| Classifier→Language Head | ✔ | ✔ | ✔ | ✔ | | ✔ | 47.9 (-7.7) |
| LSG w/o stop gradient | ✔ | ✔ | ✔ | | ✔ | | 48.3 (-7.3) |

Table 7: Ablation study using different alignment strategy to transfer language semantic knowledge to the primary model on four image datasets. *OH* denotes for *Office-Home* dataset.

| Method | FGCV Aircraft | | | Stanford Cars | | | CUB-200 | | | OH Avg. (ID) | OH Avg. (OOD) |
|---|---|---|---|---|---|---|---|---|---|---|---|
| | 15% | 30% | 50% | 15% | 30% | 50% | 15% | 30% | 50% | | |
| Vanilla (No Alignment) | 41.6 | 57.8 | 68.7 | 41.1 | 65.9 | 78.4 | 51.2 | 64.6 | 74.6 | 89.0 | 68.1 |
| Text Prototype Alignment | 51.1 | 68.8 | 76.5 | 52.4 | 73.0 | 82.6 | 56.0 | 70.1 | 76.8 | 89.0 | 68.8 |
| Contrastive Alignment | 54.2 | 70.7 | 78.3 | 54.3 | 73.8 | 83.2 | 56.7 | 69.9 | 77.1 | 89.4 | 69.3 |
| **LSG** | **55.6** | **72.0** | **79.5** | **55.4** | **75.5** | **83.8** | **57.7** | **70.6** | **77.2** | **89.6** | **71.1** |

its baseline Pseudo-Labeling by 4.28% and 4.93%, and achieves 80.36% and 52.65% accuracies on two datasets with only 10% labeled data.

For audio experiment, we utilize the Audio-MAE ViT-B model pretrained on the full *AudioSet-2M* and report its finetuning results on a class-balanced *AudioSet-20K* subset using 10%, 25% and 50% labeled audio clips. Our results is illustrated in Table 5 where we compare LSG with vanilla fine-tuning method. As a consequence, LSG achieves an average of 5.56% accuracy enhancement from the baseline, which demonstrates that guiding the audio representation by their label semantic information could also be beneficial. These results on video and audio experiments demonstrate that LSG is widely application to various modalities.

## 4.3 Analytical Experiments

**Ablation Study**. We report the accuracy of different LSG variants on *FGVC-Aircraft* with 15% labeled data (Table 6). We begin with respectively removing the regularization loss $\mathcal{L}_r$ and the alignment loss $\mathcal{L}_{align}$ from the total objective. The performance drops of 14.4% and 4.9% prove that both objectives contribute to the semantic guidance. Moving on, we investigate two variants: replacing the trainable classifier by a fixed head from language embeddings, and removing the stop gradient in $\mathcal{L}_r$. Both variants performs much worse than the current design.

Table 8: Calinski-Harabasz Indexes of the original label embeddings from pretrained language model and the GCN refined label embeddings. *OH* denotes for *Office-Home* dataset.

| Method | Aircraft | StanfordCars | CUB200 | OH |
|---|---|---|---|---|
| Original | 86.5 | 137.2 | 157.8 | 85.5 |
| **GCN refined** | **1364.1** | **1481.7** | **916.3** | **1395.2** |

**Effectiveness of the GCN model**. Here we conduct two experiments to verify the effectiveness of the GCN model on refining the label embeddings and promoting semantic knowledge transfer from LSG to the primary model. First, we compares the Calinski-Harabasz Index between the original and refined label embeddings, where higher index indicates better clustering performance. We use the node embeddings of the penultimate layer of the GCN as the refined label embeddings. As shown in Table 8, the index of the refined embeddings is significantly higher than the original ones, which demonstrates that the GCN model learns the discriminative information and improves the label embedding quality.

Second, we compare the performance of LSG with two alternative alignment strategies: Text Prototype Alignment and Contrastive Alignment. For the former, we use the average of the label embeddings as the class prototypes and minimize the cosine similarity between features and their corresponding prototypes. For the latter, we adopt supervised contrastive loss [26] between features and label embeddings. The results are shown in Table 7. We observe that both alignment strategies improve the performance of the vanilla finetuning, which demonstrates that learning semantic relation via the alignment loss is beneficial to the model training. However, the performance of LSG is still

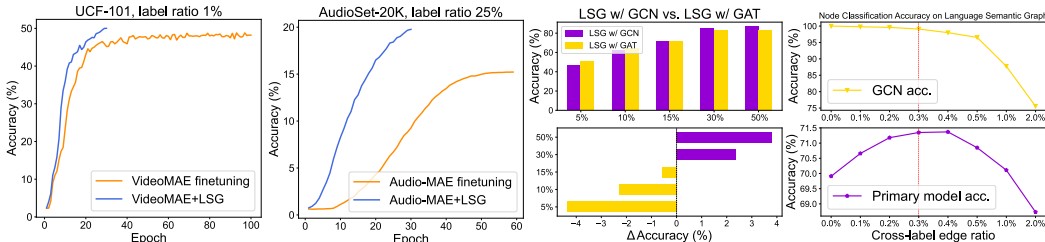

Figure 2: Analytical experiments: (a) and (b) Model accuracy on validation sets during training. (c) Comparison between GCN and GAT. (d) Effect of the threshold hyper-parameter $\tau$.

superior to the two alternatives, which indicates that the GCN model is more effective in transferring the label semantic knowledge to the primary model.

**LSG speeds up model training**. It is discovered that applying LSG to guide the representation learning encourages the model to achieve a satisfactory performance faster. The accuracy curves on validation set of LSG and its baseline in both video (*UCF-101*) and audio (*AudioSet*) training are depicted in Fig. 2(a)(b). The accuracy curve of LSG rises faster and higher than vanilla fine-tuning.

**Substitute GCN by GAT** [63]. To further investigate the effect of the graph neural network in our system, we change the implementation from GCN to Graph Attention Network (GAT), which is another wildly adopted GNN. The biggest different is that GAT automatically learns the weight between connected nodes instead of relying on manual definition as GCN does. We compare the two variants on *FGVC Aircraft* SSL with 15% labeled data and discover an interesting trend. As shown in Fig. 2(c), GAT outperforms GCN in low label regime ( labeled data $\leq 15\%$) whereas GCN is more beneficial with more labeled samples.

**Analysis of the LSG edge connection threshold** $\tau$. The hyper-parameter $\tau$ in Eq. (2) controls the topology (i.e., the number of edges that connect different labels) of the Language Semantic Graph. We study the effect of this threshold on both the GCN node classification accuracy and more importantly the primary model accuracy. We observe (in Fig. 2(d)) that as the ratio of cross-label edges decrease, the graph topology becomes more simple and GCN accuracy increases. However, due to lack of beneficial cross-label interaction, the primary model performs worse. In contrast, if the ratio is overwhelmingly high, the graph topology becomes too complicated and results in degradation in both GCN and primary model performance.

## 5   Limitation and Future Work

The proposed semantic graph is a general concept that aims to capture the high-level semantic relations between labeled concepts. In this paper, we only use the natural language descriptions of the labeled concepts and leverage the pretrained language model to construct the graph. Although using language embedding is effective in many data-efficient scenarios across various modalities, it is still limited when the labeled concepts in the task are not well described by natural language.However, the semantic relations between concepts can also be captured by other ways, such as manually determining the relationship between categories based on expert knowledge or leveraging a knowledge graph. In the future, we will explore how to combine the semantic relations from different modalities to construct a more comprehensive graph.

Meanwhile, since the proposed LSG is a simple and lightweight modification to the standard training process and does not alter the inference procedure, another promising direction would be to extend the proposed method to other data-efficient scenarios, such as semantic segmentation and object detection.

## 6   Conclusion

In this paper, we study how to exploit the semantic information in labels for improving data efficiency. We present LSG, a novel and effective data-efficient learning method that leverages Language Semantic Graph to guide the model learn semantic relations between labeled concepts. LSG consists of two parts: an auxiliary graph neural network that extracts knowledge from the semantic graph and two novel optimization objectives that transfer the knowledge to primary model. We demonstrate that LSG is applicable on image, video and audio models and brings significant performance gains to the model under Transfer Learning and Semi-Supervised Learning scenarios.

## Acknowledgement

This paper was supported by National Key R&D Program of China (No. 2021YFB3301503), the National Natural Science Foundation of China (No. 62376026), and also sponsored by Beijing Nova Program (No. 20230484296), CCF-Tencent Rhino-Bird Open Research Fund.

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
