# Supplementary Material for Language Semantic Graph Guided Data-Efficient Learning

**Wenxuan Ma**
Beijing Institute of Technology
wenxuanma@bit.edu.cn

**Shuang Li**✉
Beijing Institute of Technology
shuangli@bit.edu.cn

**Lincan Cai**
Beijing Institute of Technology
lincancai@bit.edu.cn

**Jingxuan Kang**
University of Liverpool
sgjkang3@liverpool.ac.uk

The supplementary materials include the general algorithm of our method (§ A), dataset introduction (§ B), implementation details (§ C) and additional experimental results (§ D).

## A    General Algorithm for LSG

Our method is generally applicable to various data-efficient training scenarios. Here we provide a general algorithm of LSG in Alg. 1 on a labeled dataset $\{\boldsymbol{x}_i, y_i\}_{i=1}^n$. We note that unlabeled data can also be utilized in the semi-supervised learning scenario by assigning pseudo labels via vanilla pseudo-labeling techniques as proposed in the paper or more advanced approaches.

## B    Dataset Introduction

***FGVC Aircraft*** [7] is a benchmarking dataset for aircraft visual categorization. The dataset contains 10,000 aircraft images, with 100 images for each of 100 different aircraft model variants. The data is split into 6,667 training images and 3,333 testing images.

***Stanford Cars*** [4] is a fine-grained dataset contains 16,185 images of 196 classes of cars. The data is split into 8,144 training images and 8,041 testing images, where each class has been split roughly in a 50-50 split. Classes are typically at the level of Make, Model, Year, e.g. 2012 Tesla Model S or 2012 BMW M3 coupe.

***CUB-200-2011*** [10] is the most widely-used dataset for fine-grained classification. It contains 11,788 images of 200 subcategories belonging to birds, 5,994 for training and 5,794 for testing. We only use the subcategory name for each bird to as concepts without using the fine-grained natural language descriptions.

***Office-Home*** [9] is a standard dataset for domain adaptation and domain generalization. It contains four different domains: Artistic (**Ar**), Clip Art (**Cl**), Product (**Pr**) and Real-world (**Re**). Each domain consists of roughly 4,000 images within the same 65 object categories found typically in office and home scenarios. We adopt a 80-20 split for each domain to train the model and evaluate the in-distribution performance. We adopt all the data available in the rest domain for out-of-distribution performance evaluation.

***UCF-101*** [8] is an action recognition data set of realistic action videos, collected from YouTube, having 13,320 videos from 101 action categories. The action categories can be divided into five types: human-object interaction, body-motion only, human-human interaction, playing musical instruments and sports. We adopt the first official dataset split and obtain 9,537 training videos and 3,783 test videos.

37th Conference on Neural Information Processing Systems (NeurIPS 2023).

**Algorithm 1:** Language Semantic Graph Guided Training

**Input:** Labeled Data $\{\boldsymbol{x}_i, y_i\}_{i=1}^n$; Concepts (labels in natural language) $\{\mathcal{W}_k\}_{k=1}^K$; Backbone Model $F$; Pre-trained Language Model; Prompt set $\{\mathcal{P}_q\}_{q=1}^m$; Hyper-parameters $\lambda$ and $\tau$; GCN training iteration $I_g$; Max training iteration: $I$

**Output:** Model for the Task: $C \circ F$ ($C$ is the task-specific classifier).

// Stage 1: Train a GCN on language semantic graph

1   Combine $\mathcal{P}$ with $\mathcal{W}$ to obtain input text set $T_0 = \cup_{k=1}^K \{\mathcal{P}_1 \mathcal{W}_k, \mathcal{P}_2 \mathcal{W}_k, ..., \mathcal{P}_m \mathcal{W}_k\}$;

2   Send input texts to the pre-trained language model and obtain the set of text embeddings
    $T = \cup_{k=1}^K \{\boldsymbol{t}_k^{(1)}, \boldsymbol{t}_k^{(2)}, ..., \boldsymbol{t}_k^{(m)}\}$;

3   Create the Language Semantic Graph $\mathcal{G}(\mathcal{V}, \boldsymbol{A}, \mathcal{K})$, where node embeddings comes from $T$ and adjacency matrix from Eq. (2);

4   Initialize GCN encoder $\mathcal{F}$ and classifier $\mathcal{C}$;

5   **for** $iter = 1, 2, \cdots, I_g$ **do**

6      Compute the output for every node $\boldsymbol{t}$ in the graph $\mathcal{G}$ as $\mathcal{C} \circ \mathcal{F}(\boldsymbol{t})$;

7      Update the GCN model by $\mathcal{L}_{node}$ in Eq. (4);

8   **end**

// Stage 2: Formal training stage with LSG

9   Initialize classifier $C$ and linear projector $H$;

10   Fix the weights in GCN;

11   **for** $iter = 1, 2, \cdots, I$ **do**

12      $\boldsymbol{f}_i \leftarrow F(\boldsymbol{x}_i)$;

13      $\boldsymbol{p}_i \leftarrow C(\boldsymbol{f}_i)$, $\boldsymbol{h}_i \leftarrow H(\boldsymbol{f}_i)$;

14      Compute $\mathcal{L}_{emp}(\boldsymbol{p}_i, y_i)$ by Eq. (1);

15      Connect the projected features to the graph $\mathcal{G}$ to obtain augmented graph $\mathcal{G}_{aug}(\mathcal{V}_a, \boldsymbol{A}_a, \mathcal{K}_a)$;

16      Compute the output for every projected feature $\boldsymbol{h}_i$ in th augmented graph $\mathcal{G}_{aug}$;

17      Calculate $\mathcal{L}_{align}(\boldsymbol{h}_i, y_i)$ by Eq. (5);

18      Calculate $\mathcal{L}_r(\boldsymbol{h}_i, y_i)$ by Eq. (6);

19      $\mathcal{L} \leftarrow \mathcal{L}_{emp} + \lambda \mathcal{L}_{align} + \mu \mathcal{L}_r$;

20      Update model by $\mathcal{L}$;

21   **end**

---

***HMDB51*** [5] dataset is a large collection of realistic videos from various sources, including movies and web videos. The dataset is composed of 6,766 video clips from 51 action categories (such as "jump", "kiss" and "laugh"), with each category containing at least 101 clips. We adopt the official 70-30 split.

***AudioSet*** [2] is an audio event dataset, which consists of over 2M human-annotated 10-second video clips collected from YouTube. It contains 527 categories, in which an audio clip are often annotated to multiple categories. All the videos are split into Evaluation/Balanced-Train/Unbalanced-Train set. The balanced training set is referred to *AudioSet-20K* and the complete training set is *AudioSet-2M*. We adopt Audio-MAE pre-trained on the complete *AudioSet-2M* set and fine-tune the model on *AudioSet-20K*.

## C   Implementation Details

**Text embeddings from Pre-trained Language Model.** We adopt the modified version of 20 prompts provided in CLIP, which is adding "This is" before the original prompt to make the sentences more complete. For the text embeddings, we directly use the output embedding corresponding to the concept instead of the [cls] embedding or [eos] embedding. If the concept is divided into several tokens by the tokenizer, we simply take the average of the corresponding output. We find that using the text embeddings obtained in this way generally yields better results than using [cls] token embedding or [eos] token embedding. We think it is because special tokens in pre-trained language models focus more on representing the entire sentences rather than the specific object, making the output embeddings more likely to from "prompt" cluster.

**Training Details.** We use SGD with momentum 0.9 as optimizer for training GCN, and the learning rate is set as 1e-3. On three fine-grained image classification dataset, we set the mini-batch size as 24 for both labeled and unlabeled data and adopt the step decay learning rate schedular following [11]. We only apply strong data augmentations [1] in semi-supervised learning tasks. On Office-Home dataset, we follow the settings within the domain generalization codebase [3], using mini-batch size of 64 and train 250 iterations for 40 epochs. For both video and audio experiments with masked autoencoder as pre-trained model, we fine-tune the model using four gpus on a single node instead of the configurations from official scripts due to our resource constraint. Such a change will slightly lower the model performance.

# D   Additional Experiments

## D.1   More Results on Data-Efficient Image Classification

**Data-Efficient Training on Large Dataset.** Following Co-tuning [12], we evaluate LSG on a larger-scaled image classification dataset COCO-70 (DenseNet-121) and report the results in table below. Our method is superior than Co-tuning under every labeling ratio on this large-scale benchmark, implying a boarder application range of LSG.

Table 1: Classification accuracy in large-scale COCO-70 dataset (Pre-trained DenseNet-121).

| Method | 15% | 30% | 50% | 100% |
|---|---|---|---|---|
| Fine-tuning | 76.60 | 80.15 | 82.50 | 84.41 |
| Co-tuning | 77.64 | 81.19 | 83.43 | 85.65 |
| **LSG** | **79.50** | **82.33** | **84.14** | **86.11** |

**Single Domain Generalization on PACS.** Here we show that LSG is still semantic meaningful and works well when the category number is small. We evaluate single domain generalization on another classical benchmark PACS that has only 7 categories. The results are shown in table 2. Similar to the experiments on Office-Home, We find that LSG also brings significant performance improvement compared to vanilla empirical risk minimization.

Table 2: Single domain generalization on *PACS* that only has 7 categories.

| Backbone | Method | Source:P | | | | Source:A | | | | Source:C | | | | Source:S | | | | Avg. (ID) | Avg. (OOD) |
|---|---|---|---|---|---|---|---|---|---|---|---|---|---|---|---|---|---|---|---|
| | | P | A | C | S | A | P | C | S | C | P | A | S | S | P | A | C | | |
| ConvNext-S | ERM | 100 | 75.6 | 37.0 | 31.6 | 99.5 | 99.1 | 76.2 | 82.0 | 99.3 | 90.2 | 89.8 | 77.5 | 99.1 | 44.4 | 56.1 | 75.8 | 99.5 | 69.6 |
| | **LSG** | 100 | **85.4** | **48.4** | **52.7** | 99.1 | **99.3** | **80.8** | **86.0** | 99.8 | **97.2** | **90.3** | **82.1** | 99.2 | **56.5** | **64.9** | 74.4 | 99.5 | **76.5** |

Table 3: Classification accuracy with noisy labels (Pre-trained ResNet-50).

| Method | *FGCV Aircraft* | | | | *Stanford Cars* | | | | *CUB-200* | | | |
|---|---|---|---|---|---|---|---|---|---|---|---|---|
| | 15% | 30% | 50% | 100% | 15% | 30% | 50% | 100% | 15% | 30% | 50% | 100% |
| Fine-tuning (clean) | 41.6 | 57.8 | 68.7 | 80.2 | 41.1 | 65.9 | 78.4 | 87.8 | 6 | 72.3 | 76.3 | 78.7 |
| **LSG** (clean) | **55.6** | **72.0** | **79.5** | **86.7** | **55.4** | **75.5** | **83.8** | **90.7** | **57.7** | 70.6 | **77.5** | **82.2** |
| Fine-tuning (20% noise) | 27.2 | 38.8 | 50.7 | 62.6 | 21.7 | 43.6 | 58.1 | 71.8 | 36.2 | 50.1 | 57.4 | 69.7 |
| **LSG** (20% noise) | **41.5** | **54.8** | **61.9** | **72.6** | **36.3** | **54.3** | **67.9** | **80.8** | **42.1** | **57.3** | **64.2** | **73.8** |
| Fine-tuning (40% noise) | 15.7 | 20.7 | 33.7 | 44.3 | 13.9 | 27.8 | 37.7 | 51.3 | 23.0 | 35.2 | 44.0 | 52.6 |
| **LSG** (40% noise) | **27.5** | **35.6** | **44.7** | **52.8** | **23.4** | **38.0** | **46.4** | **58.6** | **27.9** | **39.5** | **47.2** | **56.9** |

**Learning with Noisy Labels.** To demonstrate the wide applicability of our method, we investigate an alternative scenario where labels in the training data contain noise. We randomly add label noise to the clean datasets and compare LSG with fine-tuning baseline. The corresponding results under different noise and labeling ratio on three benchmarks are shown in table 3, in which 10% noise means that 10% of the training data are randomly assigned to false labels.

The results show that both methods are influenced by the label noise and their performance degradation is observed. Still, LSG outperforms fine-tuning baseline significantly on each noise level, and achieves relatively less accuracy degradation from clean label scenarios. The possible reason behind such robustness is that LSG regularizes the image feature space to maintain the category relationships, which prevents the feature encoder learns heavily distorted feature space by overfitting to the noisy data. Moreover, methods that identify and remove noisy labels can be integrated to LSG on these scenarios to improve its performance.

Table 4: Accuracy of CLIP ViT-B/16 with tunable language head and CLIP language encoder.

| Method | FGCV Aircraft | | | | Stanford Cars | | | | CUB-200 | | | |
|---|---|---|---|---|---|---|---|---|---|---|---|---|
| | 15% | 30% | 50% | 100% | 15% | 30% | 50% | 100% | 15% | 30% | 50% | 100% |
| CLIP Fine-tuning | 39.4 | 50.0 | 56.7 | 65.1 | 69.2 | 77.8 | 80.5 | 83.7 | 60.7 | 71.0 | 74.9 | 78.3 |
| + language head (Tunable) | 45.1 | 54.9 | 60.1 | 68.1 | 74.3 | 80.0 | 82.7 | 86.5 | 67.4 | 74.5 | 77.3 | 82.1 |
| **LSG** | 48.9 | **58.6** | **65.0** | **74.5** | **79.4** | 83.2 | **86.1** | **90.1** | 70.2 | **78.4** | **81.9** | **85.4** |
| **LSG w/ CLIP**$_{\text{text}}$ | **49.1** | **58.6** | 64.5 | 74.4 | 79.2 | **83.4** | 85.9 | 89.6 | **70.6** | 77.9 | 81.8 | 85.1 |

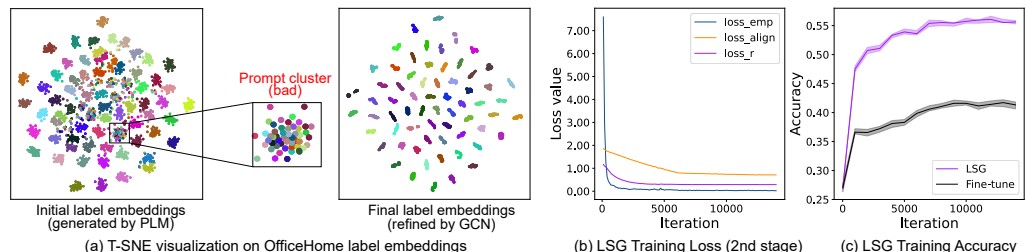

Figure 1: Additional analytical experiments: (a)T-SNE visualization of the original and refined label embeddings of *OfficeHome* datasets; (b) Three loss curves during the LSG primary training stage, showing that the training process of LSG is steady; (c) The variation of training accuracies on *Aircraft-15% sample* for five runs with different seed.

This experiment shows the potential of using LSG on scenarios with low quality labels. Still, we acknowledge that there are other scenarios where label quality is not satisfying, and we will leave it for future study.

## D.2 Additional Analysis

. **Visualization of the Refined Label Embeddings.** We visualize the original label embeddings and the refined label embeddings from GCN penultimate layer in Fig. 1(a). We can see that in the original label embeddings, some poorly-crafted prompt template may generate "prompt cluster" in embedding space due to its dominance influence on the label embedding over the concept that we want to distinguish. On the other hand, with the help of GCN, the refined label embeddings are more compact and the intra-class distance is smaller than the original label embeddings. This indicates that the GCN can refine the label embeddings to be more discriminative and compact, which is beneficial for the classification task.

**Comparison to using CLIP language encoder.** We also compare the performance of LSG with using CLIP language encoder instead of the pre-trained language model. The results are shown in table 4. Here we consider two new variants: (1) CLIP Fine-tuning + language head (Tunable) refers to the stronger baseline by making the language embedding initialized classifier tunable. We draw the same conclusion as in ELEVATER [6] that the tunable head achieves better performance than random initialization. Yet, our method still outperforms this language-augmented baseline on all the tasks with a great margin. (2) LSG w/ CLIP$_{\text{text}}$. We use the CLIP$_{\text{text}}$ as the language encoder in LSG. We can see that LSG w/ CLIP$_{\text{text}}$ achieves similar performance as LSG, which indicates that both the pre-trained language model BERT and CLIP$_{\text{text}}$ are suitable for LSG.

**The Training Stability**. We show the training curves of the three losses in the primary training stage of LSG in Fig. 1(b). We can see that the training process is steady and the three losses are all decreasing during the training process. Also, we plot the variance of training accuracies on *Aircraft-15% sample* for five runs with different seed in Fig. 1(c). We can see that the variance is small and similar to the vanilla fine-tuning, which indicates that introducing the extra supervision from language semantics does not increase the variance of the training process.