# OpenReview forum: "Language Semantic Graph Guided Data-Efficient Learning"
_NeurIPS.cc/2023/Conference — NeurIPS 2023 poster_

### Official Review · Reviewer_A9AK · 2023-06-29

**Soundness:** 3 good
**Presentation:** 3 good
**Contribution:** 3 good
**Rating:** 6
**Confidence:** 4

**Summary:**

This paper proposed a general framework for exploiting semantic information within labels in classification tasks to improve the performance of deep nueral networks. The framework is both task-agnostic and model-agnostic, making it applicable to a wide range of classification tasks and modalities. The framework first concatenates label concepts/descriptions with prompting templates, and feed these templates into a frozen language models to generate embeddings. These embeddings are subsequently used to construct a semantic graph, which is trained using a graph convolutional network (GCN). When applying the semantic graph for guiding the training process, two regularization objective is added based on the semantic grpah in order to enhance the model with label semantics. The goal of the whole framework is similar to the prompt learning in NLP, while its capability of applying to tasks in other modality improve its novelty and contributions. In experiments, the authors show that their proposed framework successfully outperforms many other baselines based on transfer learning, semi-supervised learning and data augmentation.

**Strengths:**

1. The paper itself is clearly written and presented, enabling readers to understand the proposed framework easily.
2. The proposed framework for incorporating semantic information of labels for tasks in various modalities is novel and widely applicable.
3. From the experiment section, the performance of the proposed framework significantly outperforms several baselines based on transfer learning, semi-supervised learning, and data augmentation.

**Weaknesses:**

The proposed framework requires that the target task be a classification task and that each class has explicit and rich semantics. If a class does not have an explicit semantic meaning, the LSG may not work. This requirement limits the applicability of the LSG framework to many classification tasks.

**Questions:**

My questions are mainly about the generalization ability of the proposed framework:


1. Can you provide more examples of prompt templates used for generating embeddings?

2. What is the performace of data on some large-scale datasets (i.e. when the data source is rich)? In Co-tuning's paper, they also conducted experiments COCO-70, which is considered a large-scale dataset with 1k samples per class.

3. What is the ituition behind adding sample-sample interaction in the augmented graph (i.e. the matrix M). Does it bring any benefit to LSG?


4. I noticed that all the tasks used in the experiments have at least 50 categories. I am curious if the LSG framework can be applied to binary classification tasks or tasks with a small number of categories (<10). In my opinion, it may be difficult to apply LSG to binary classification because the descriptions of the labels are generally two sides of one statement. For tasks with fewer than 10 categories, it may be challenging to train a GCN. It would be great if the authors could have some disscussion about these tasks.



**Limitations:**

The authors did not discuss limitations of the work. I do not see any potential negative societal impact.

---

> ### Author Rebuttal · Authors · 2023-08-09
>
> We thank you for your positive feedback and comments on our submission. Please find our responses your questions below.
>
> **Q1. Scenarios where label semantics are weak.**
>
> A1. Thanks for your insightful comment. It is a solid concern, as there exists certain scenarios like defect detection where different classes are merely represented as "defect 1", "defect 2" and etc. In this case, the label embedding obtained from pre-trained language model is less meaningful.
>
> However, we argue that **the idea of SG (semantic graph) guided knowledge transfer may still be applicable if we can construct a graph reflecting the relationship between these classes without the help of the language model**. For example, if we know that some types of the defects are caused by similar machine fault, we can use these relations to build graphs like stochastic block model, and still apply the proposed method to guide the primary model with the graph topology.
>
> To demonstrate the wide applicability of our method, we investigate an alternative scenario where labels in the training data contain noise. It shares certain similarity with the situation you mentioned, as part of the training labels can no longer reflect the true nature of the samples. We randomly add label noise to the clean datasets and compare LSG with fine-tuning baseline.
>
> The table below shows that LSG still outperforms fine-tuning baseline significantly, and results in less accuracy degradation from clean label scenarios. (20% noise means that 20% of the training data are randomly assigned to false labels.)
>
> | Variants| Air-15%  | Air-30%  | Air-50%  | Air-100% | Car-15%  | Car-30%  | Car-50%  | Car-100% | CUB-15%  | CUB-30%  | CUB-50%  | CUB-100% |
> | -- | -- |-- |-- |--|--|--| --| -- |--|--|--|--|
> | Fine-tuning  clean    | 41.6     | 57.8     | 68.7     | 80.2     | 41.1     | 65.9     | 78.4     | 87.8     | 51.2     | 64.6     | 74.6     | 81.8     |
> | **LSG clean**         | **55.6** | **72.0** | **79.5** | **86.7** | **55.4** | **75.5** | **83.8** | **90.7** | **57.7** | **70.6** | **77.5** | **82.2** |
> | Fine-tuning 20% noise | 27.2     | 38.8     | 50.7     | 62.6     | 21.7     | 43.6     | 58.1     | 71.8     | 36.2     | 50.1     | 57.4     | 69.7     |
> | **LSG 20% noise**     | **41.5** | **54.8** | **61.9** | **72.6** | **36.3** | **54.3** | **67.9** | **80.8** | **42.1** | **57.3** | **64.2** | **73.8** |
> | Fine-tuning 40% noise | 15.7     | 20.7     | 33.7     | 44.3     | 13.9     | 27.8     | 37.7     | 51.3     | 23.0     | 35.2     | 44.0     | 52.6     |
> | **LSG 40% noise**     | **27.5** | **35.6** | **44.7** | **52.8** | **23.4** | **38.0** | **46.4** | **58.6** | **27.9** | **39.5** | **47.2** | **56.9** |
>
> In summary, we acknowledge that this is an important future direction for LSG  and argue that our proposed method has the potential to adapt to these more challenging scenarios. We will add discussion on this issue in our paper to motivates future works.
>
> **Q2. Examples for prompt templates.**
>
> A2. We adopt the first twenty original hand-crafted prompt templates provided by CLIP, which includes:
> - "This is a bright photo of a {}",
> - "This is a bad photo of a {}",
> - "This is a photo of many {}",
> - "This is a sculpture of a {}",
> - "This is a tattoo of a {}" etc.
>
> We discover that this set of prompt templates works well even for the video and audio task, and thus do not manually design new templates.
>
> **Q3. Performance on large-scale COCO-70 dataset.**
>
> A3. Following Co-tuning, we evaluate LSG on COCO-70 (DenseNet-121) and report the results in table below. **Our method is superior than Co-tuning under every labeling ratio on this large-scale benchmark**, implying a boarder application range of LSG.
> |Method| Air-15% |Air-30%|Air-50%|Air-100%|
> | -- | :--: | :--: | :--: | :--: |
> |Fine-tune| 76.60|80.15|82.50|84.41|
> |Co-tuning|77.64|81.19|83.43|85.65|
> | LSG| **79.50** | **82.33** | **84.14** | **86.11** |
>
> **Q4. About sample-sample interaction.**
>
> A4. The interaction described in adjacency matrix $M$ creates additional path for knowledge transfer for each image feature, which brings impact to the optimization of both losses $\mathcal L_{align}$ and $\mathcal L_{r}$.
>
> - For $\mathcal L_{align}$, it means that each image features are affected not only by label embeddings but also image features from the same class. To guarantee correct node classification under such influence, the consistency between image features are promoted.
> - On the other hand, the existence of these sample-sample interaction means that the image feature after GCN refinement (i.e., $\mathcal F(\tilde{h})$) naturally incorporates features from other images and consequently have richer information. Therefore, $\mathcal L_{r}$ obtains better targets for the original feature $h$ to learn. The following ablation study, in which we replace $M$ by an identity matrix $I$ , supports our argument.
>
> | Variants| *Aircraft-15%* | *Aircraft-30%* | *Aircraft-50%* | *Aircraft-100%* |
> | -- | :--: | :--: | :--: | :--: |
> | LSG w/ $I$ | 55.0 | 71.4 | 79.1 | 86.1 |
> | LSG w/ $M$ | **55.6** | **72.0** | **79.5** | **86.7** |
>
> **Q5. LSG on tasks with fewer categories.**
>
> A5. Good question. We must say that it is merely a coincidence that all the tasks we evaluate in the paper involve many categories. Here we show that **LSG is still semantic meaningful and works well when the category number is small**. We evaluate single domain generalization on another classical benchmark PACS that has only 7 categories (see Table C in pdf). We find surprisingly that LSG brings more significant performance improvement than on OfficeHome.
>
> As for the case of binary classification, it is correct that the category relationship captured by the semantic graph is no longer useful, since the only thing that matters then is to discriminate between the two classes. However, the label embeddings provided by the pre-trained language model may still be useful to prevent feature distortion.

---

> > ### Comment · Reviewer_A9AK · 2023-08-18
> > **Thank you for the response.**
> >
> > Thank you for the detailed rebuttal, It makes me feel clearer toward this paper. After reviewing all other reviewers opinions and the corresponding response from the authors, I decide to remain my current score.

---

> > > ### Author Response · Authors · 2023-08-18
> > > **Thank you**
> > >
> > > We thank you again for your valuable time and feedback.

---

### Official Review · Reviewer_3r7Y · 2023-06-30

**Soundness:** 2 fair
**Presentation:** 1 poor
**Contribution:** 2 fair
**Rating:** 4
**Confidence:** 4

**Summary:**

This paper employ a language semantic graph to cature the relationship among different class, with the hope to alleviate the requirement of extensive training data and, in particular, human supervision. Generally, this paper first build the language semantic graph  with the pre-trained language models. After that, the paper introduces two additional losses $L_align$ and $L_r$ that exploit the discarded label
 semantic information for complement.

**Strengths:**

I have carefully read this papar some times, and I'm interested in injecting knowledge to the model for effective learning. The strengths can be describued as follow:

1. The idea is meaningful and interesting that employing prior knowledge  for effective learning, and the proposed method is novel, at least I've not read some related paper.

2. The experiment results  are effective compred with baseline models.

**Weaknesses:**


1. The author not provide the code, and I'm very confued on the key section.

**Questions:**

1. The final loss function in eq.(7) don't include the loss L_node, and what is the influce of L_node?

2. The language semantic graph is built with the embedding of label, so why you not using the embedding to capture the smantic relathionship among labels rather than using a discrete semantic graph?

3. I'm confused on m the size of the prompt set, what is the prompt set?

4.The section 3.2 are very confused for me, why the two proposed loss can help  data effective learning.  The author doesn't give a detail description for eq(5) and eq(6). And I'm very confused on why need the proposed two align loss, and the expective result of the proposed align loss.




**Limitations:**

The developed method are depend on pre-trained language model, and sometimes the labels in a task maybe not have an effective embedding with the pretrained language models.

---

> ### Author Rebuttal · Authors · 2023-08-09
>
> Thanks for your questions. We'll do our best to explain our method here and will thoroughly revise the paper to improve its clearity.
>
> **Q1. Code for better understanding.**
>
> A1. Our code is now provided to AC, please refer to it for better understanding.
>
> **Q2. About the effect of loss $\mathcal L _{node}$.**
>
> A2. The proposed LSG consists of two stages. In the first stage, we train a GCN on the language semantic graph $\mathcal G$. In the second stage, we train the primary model $F$  (i.e., ResNet) with the aid of the trained GCN. The loss $\mathcal L_{node}$ is only used for training GCN in the first stage, thus it is not appeared in eq.(7) for the second stage.
>
> $\mathcal L_{node}$ optimizes the GCN to correctly classify each node on the semantic graph $\mathcal G$. By doing this, each node processed by the GCN aggregates discriminative and more comprehensive semantic information from neighboring nodes (see T-SNE visualization of the label embeddings in figure 3(a)).
>
> The trained GCN is then fixed as a graph processor and deployed on the augmented semantic graph $\mathcal G_{aug}$ to pass semantic information to new nodes in the second stage. More specifically, eq. (5) and eq. (6)  leverage the ($\mathcal L_{node}$) trained GCN to guide the feature learning of the primary model, which will be explained later.
>
> **Q3. About the prompt set.**
>
> A3. The prompt set is a collection of hand-crafted prompts where each one can complete a given concept into sentence. For example, one prompt can be 'This is a photo of {}' or 'This is a sculpture of {}' where we can replace '{}' by categories. The size of the prompt set $m$ determines the number of different sentences we can create for each concept, which consequently determines the number of text embeddings. (That is why we have total $|T|=mK$ embeddings for $K$ classes.) Via prompt set, we can obtain multiple different embeddings corresponding to the same category. The advantage of such diversity is fully utilized in the constructed semantic graph that will be explained next.
>
> Table below shows how $m$ affects the accuracy. We choose $m=20$ to achieve a balance between performance and cost.
>
> ||$m=$5 |10|20|40|80|
> |--|:--:|:--:|:--:|:--:|:--:|
> |Air-15%|54.8|55.0|55.6|55.7|55.7|
>
> **Q4. Using a semantic graph instead of the original label embeddings**
>
> A4. The proposed semantic graph provides a better supervision than using the original embeddings directly does. The reason is three fold.
>
> - First, the semantic graph is built upon label embeddings (i.e., $\mathcal K^{(0)}=\mathcal K$), and thus includes all the information that label embeddings have.
> - Second, information can be passed on among neighboring nodes to produce more discriminative and comprehensive node embeddings (see T-SNE), and further be transferred to new node of image features through new links. Therefore, our semantic graph, with the aid of the GCN trained by $\mathcal L_{node}$, allows the image features capture semantics from label embeddings.
> - Third, we empirically verify that using semantic graph is more effective than simply supervise primary model features by label embeddings using different classical alignment strategies. As shown in table below, both the prototype alignment and contrastive alignment method that only leverage initial label embeddings perform worse than LSG.
>
> ||Air-15%|Air-30%|Air-50%|Air-100%|
> |--|:--:|:--:|:--:|:--:|
> |Prototype align.|51.1|68.8|76.5|84.6|
> |Contrastive align.|54.2|70.7|78.3|85.1|
> |LSG|**55.6**|**72.0**|**79.5**|**86.7**|
>
> **Q5. About Section 3.2**.
>
> A5. This section describes the process of injecting label semantic knowledge into the primary model. The motivation behind it is that common practice of cross-entropy loss turns labels into one-hot vectors, during which the semantic relationship between categories are erased. (All these one-hot vectors are perfectly perpendicular to each other.) Thus, to introduce the lost semantic information into the network makes the better use of the training data and promote data-efficient learning. Take vision model as an example, our goal is to align the image features towards the semantic space of label embeddings provided by the PLM, in which the relationship between categories are better reflected than in the one-hot space.
>
> The alignment is conducted both implicitly (via eq. (5)) and explicitly (via eq. (6)), and their effect are explained as follows:
>
> - Eq. (5) is the node classification loss on new nodes from image features $h$. Since that the GCN trained by $\mathcal L_{node}$ in the first stage can already classify label embeddings (original nodes) correctly and it is now set fixed, lowering the cost of eq. (5) forces the image features become similar to their corresponding label embeddings, achieving implicit feature alignment.
> - Eq. (6) computes the $l_2$ distance between original image features and the new features encoded by the GCN, therefore explicitly pushes the former one towards the latter one (which is detached). Note that the new features have aggregated label embeddings from their neighbors in the augmented semantic graph, thus are desirable representations for the vision model. Please refer to the algorithm provided in the pdf for the complete training process.
>
> The ablation study conducted in the paper shows the effectiveness of both losses, and we present it here.
>
> ||LSG w/o $\mathcal L_{align},\mathcal L_r$|LSG w/o $\mathcal L_{align}$|LSG w/o $\mathcal L_r$|LSG|
> |--|:--:|:--:|:--:|:--:|
> |Air-15%|41.6|44.2|50.7|**55.6**|
>
> **Q6. On tasks without effective language embeddings.**
>
> A6. This is a solid concern. Although LSG cannot be directly applied to these tasks, we argue that our idea of semantic graph guided knowledge transfer is applicable if we can construct a graph reflecting the relationship between classes without PLM. In this way, our proposed method can once again guide the model to learn the category relations.
>
> We hope these responses address your concerns.

---

> > ### Comment · Reviewer_3r7Y · 2023-08-15
> >
> > Thanks for your reponse.  Some concern have been solved.
> >
> > I don't know how can I get code from AC.

---

> > > ### Author Response · Authors · 2023-08-15
> > > **We have posted a new official comment and asked for permission to post code link**
> > >
> > > According to the official notification, we cannot post our code link here. (In notification) "All the texts you post (rebuttal, discussion and PDF) should not contain any links to external pages. If you were asked by the reviewers to provide code, please send an anonymized link to the AC in a separate comment."
> > >
> > > To address this issue, we have posted a new official comment visible to all reviewers and the AC at the top, and asked for AC's permission to post our code link there. We believe you will be informed once we get the permission.

---

> > > > ### Comment · Reviewer_3r7Y · 2023-08-15
> > > >
> > > > Thanks for your further response, and can you describe the detail of Prototype align. and Contrastive align.  Further, I think providing experiment results of these two methods on all the other datasets is help to further improve the quality of papers.

---

> > > > > ### Author Response · Authors · 2023-08-16
> > > > > **Code link is now available and results of the two variants on all datasets**
> > > > >
> > > > > Thanks for your valuable feedback that helps to improve our work, and we would like to assure you that all our results can be reproduced by the code we provided, if that is what you concerned.
> > > > >
> > > > > Here we make further clarification of the two ablation variants we used:
> > > > > - prototype alignment: each class prototype is calculated by the average of the label embeddings from this class. $c_k=\frac{1}{m}\sum_i{t_k^{(i)}}$, and the prototype alignment loss for each image feature $h$ with label $y$ is the cross-entropy loss $l_{pa} = -\log \frac{e^{h\cdot  c_y}}{\sum_k e^{h\cdot c_k}}$ after applying $l_2$ normalization on both features and prototypes.
> > > > > - contrastive alignment: label embeddings are divided into $K$ subsets according to their embedded label, i.e., $[t_k^{(1)},...,t_k^{(m)}]$ for label $k$. For each image feature $h$ with label $y$, the contrastive alignment loss is a supervised contrastive loss [1] regarding embeddings in subset $y$ as positive pairs and other embeddings as negative pairs.
> > > > >
> > > > > [1] Supervised Contrastive Learning.
> > > > >
> > > > > We show the full results of these two variants on *Aircraft*, *StanfordCars*, *CUB200* for fine-tuning and *OfficeHome* for domain generalization.
> > > > > |Method|Air-15%|Air-30%|Air-50%|Air-100%| Car-15% | Car-30% | Car-50% | Car-100% | CUB-15% | CUB-30% | CUB-50% | CUB-100% |
> > > > > |--|:--:|:--:|:--:|:--:|:--:|:--:|:--:|:--:|:--:|:--:|:--:|:--:|
> > > > > |Prototype Alignment|51.1|68.8|76.5|84.6| 52.4    | 73.0    | 82.6    | 89.5     | 56.0    | 70.1    | 76.8    | 81.9     |
> > > > > |Contrastive Alignment|   54.2   |   70.7   |   78.3   |   85.1   |   54.3   |   73.8   |   83.2   |   90.1   |   56.7   |   69.9   |   77.1   |   82.1   |
> > > > > |**LSG**| **55.6** | **72.0** |**79.5**|**86.7**| **55.4** | **75.5** | **83.8** | **90.7** | **57.7** | **70.6** | **77.2** | **82.2** |
> > > > >
> > > > > | Source domain         |    Ar    |    Ar    |    Ar    |    Ar    |    Cl    |    Cl    |    Cl    |    Cl    |    Pr    |    Pr    |    Pr    |    Pr    |Rw|Rw|Rw|Rw| Avg. ID| Avg. OOD|
> > > > > | --------------------- | :------: | :------: | :------: | :------: | :------: | :------: | :------: | :------: | :------: | :------: | :------: | :------: | :------: | :------: | :------: | :------: | :------: | :------: |
> > > > > | Target domain         |    **Ar**    |    Cl    |    Pr    |    Rw    |    **Cl**    |    Ar    |    Pr    |    Rw    |    **Pr**    |    Ar    |    Cl    |    Rw    |**Rw**|Ar|Cl|Pr| Avg. ID| Avg. OOD|
> > > > > | Prototype Alignment   |   85.0   |   56.9   |   71.3   |   78.1   |   86.2   |   69.5   |   72.9   |   75.9   |   94.3   |   61.9   |   50.4   |   80.6   |90.7|71.8|55.9|81.0|89.0|68.8|
> > > > > | Contrastive Alignment |   85.6   |   57.1   |   71.6   |   78.5   | **86.3** |   69.9   |   73.0   |   76.7   |   95.0   |   62.5   |   52.1   |   80.7   |90.8|72.2|56.2|81.4|89.4|69.3|
> > > > > | **LSG**               | **85.8** | **57.7** | **74.0** | **79.9** | **86.3** | **71.7** | **75.4** | **77.8** | **95.1** | **65.8** | **54.7** | **82.1** |**91.2**|**73.8**|**58.0**|**82.3**|**89.6**|**71.1**|
> > > > >
> > > > > We observe that LSG consistently outperforms these two variants. These results will be added to the appendix of our paper.

---

> > > > > > ### Comment · Reviewer_3r7Y · 2023-08-18
> > > > > >
> > > > > > Thanks for your detailed response. In my opinion, the aim of the proposed graph method is to capture the semantic relationship between labels and discriminative information from labels. I have raised my score from 3 to 4.

---

> > > > > > > ### Author Response · Authors · 2023-08-18
> > > > > > > **Thank you**
> > > > > > >
> > > > > > > We're glad to hear that your concerns have been addressed.

---

> > > > ### Comment · Area_Chair_Yazq · 2023-08-15
> > > > **Code link**
> > > >
> > > > The authors sent me the anonymized code link. You can find it here:
> > > >
> > > > https://transfer.sh/Woy9yv4SYN/lsg_code.zip

---

### Official Review · Reviewer_pDMD · 2023-07-06

**Soundness:** 3 good
**Presentation:** 3 good
**Contribution:** 3 good
**Rating:** 6
**Confidence:** 4

**Summary:**

The paper addresses the importance of labels’ semantic meanings when training models. First, the framework constructs an LSG graph. Node features are text embeddings generated by language models, and the similarity matrix constructs edges. After that, a GCN is trained to aggregate node features of the LSG with the node classification loss. The GCN will be used for data-efficient learning afterward (regularization loss and alignment loss).

**Strengths:**

* The motivation and proposed method is sound.
* The details are well illustrated and are easy to understand.
* Experiments cover a variety of tasks and well support the claims. Ablation studies also show the effectiveness of each component.


**Weaknesses:**

* Since the LSG is the critical component of the proposed framework, the quality analysis of LSG is preferred. For example, a visualization of LSG showing how well it grasps the semantics of labels.
* The notion should be clearer. The F, C and $\mathcal{C}$, $\mathcal{F}$ are confusing when first reading. I would suggest using subscripts to distinguish them.


**Questions:**

* How to make the inference? Is GCN used in the inference or only the primary model is used?
* I am curious about the stability of the training. The augmented graph contains a batch of inputs, possibly introducing interference between samples after GCN. Will this affect the training quality/stability? For example, will the training be affected much if we change the batch size or random seed?
* Is the primary model feature encoder the same as the model used to generate node embeddings? I.e., are “LM” and “F” in the figure 1 the same?


**Limitations:**

Authors do not include limitations. Some limitations I can think of:
1. The current structure may have much more training burden than vanilla fine-tuning.
2. The method is only applicable for classification methods.
3. Authors could also try experiments on text tasks.

---

> ### Author Rebuttal · Authors · 2023-08-09
>
> We thank you for your positive feedback and comments on our submission. Please find our responses your questions below.
>
> **Q1. Quality analysis of LSG.**
>
> A1. We provide two analysis to thoroughly evaluate LSG.
>
> - We show the T-SNE visualization of the initial node embeddings and the GCN refined node embeddings based on LSG (see Figure 3(a)). We can see that in the initial embedding space there exists a few "prompt clusters" where label embeddings from different categories are clustered because of some poorly-crafted prompt templates. On the other hand, **the GCN refined embedding space consists of cleaner and more compact clusters, demonstrating that LSG improves the quality of the label embeddings** (which will in turn benefit the primary network via loss $\mathcal L_{align}$ and $\mathcal{L}_r$).
> - Secondly, please recall that loss $\mathcal{L}_r$ utilizes image feature refined by LSG (i.e., $\mathcal{F}(\tilde{h})$) as target to supervise the original one. Therefore, the improvement brought by $\mathcal{L}_r$ gives credit to the rich semantics captured in LSG.
> |Variant |Air-15%|Car-15%|CUB-15%|
> |--|:--:|:--:|:--:|
> |LSG w/o $\mathcal L_r$ | 44.2 |43.3|53.9|
> |LSG | **55.6** | **55.4** | **57.7**|
>
> **Q2. Change of notations.**
>
> A2. Thanks for your suggestion. We change the notation $\mathcal{C}$ and $\mathcal{F}$ to $\mathcal{C}_g$ and $\mathcal{F}_g$ to better represent the classifier and the encoder of the GCN, and we keep the notation of $C$ and $F$ for the primary network for simplicity.
>
> **Q3. About model inference.**
>
> A3. The GCN (and correspondingly the projector) is not used during inference and the classification result is still produced by the classifier. For this reason, **no extra computational cost is needed in inference time**.
>
> **Q4. Training  stability.**
>
> A4. To test the stability of our method, we conduct three experiments.
>
> - We plot the three loss curves for the primary training stage in Figure 3(b) and show that all the losses decrease steadily and converges.
> - We run LSG five times with different random seed. The accuracy curve in Figure 3(c) shows that LSG maintains a small accuracy variance throughout the training process.
> - As demonstrated in the following table, batch size adjustment within a certain range has minimal effect on the performance. In fact, the sample-sample interaction within a mini-batch, which is also mentioned by reviewer A9AK, is found to be beneficial to the performance. The reason is that such interaction allows each image feature to aggregate more information.
>
> The table below shows the performance of LSG with different batch size.
> | Batch Size| StanfordCars-15% | StanfordCars-30% | StanfordCars-50% | StanfordCars-100% |
> | -- | :--: | :--: | :--: | :--: |
> | 24 | 55.3 | 75.4 | **84.1** | 90.7 |
> | 48 (in paper) | 55.4 | **75.5** |83.8|90.7|
> | 64 | **55.5** | 75.2| 83.8 | **90.9** |
>
> The table below shows comparison between enabling sample-sample interaction (w/ $M$) or not (w/ identity matrix $I$).
> | Variants| Aircraft-15% | Aircraft-30% | Aircraft-50% | Aircraft-100% |
> | -- | :--: | :--: | :--: | :--: |
> | LSG w/ $I$ | 55.0 | 71.4 | 79.1 | 86.1 |
> | LSG w/ $M$ | **55.6** | **72.0** | **79.5** | **86.7** |
>
> **Q5. LM is different from $F$.**
>
> A5. "LM" is a pre-trained language model such as BERT that encodes sentences with concepts in to text embeddings (node embeddings). The primary model feature encoder $F$ is a model from a non-language modality such as ResNet. We will revise our paper to clarify the meaning of "LM" in figure 1.
>
> **Q6. About the extra training burden compared to vanilla fine-tuning.**
>
> A6. The proposed LSG does not bring too much training burden due to the following reasons.
>
> - Firstly, both GCN and the projector added to the network are lightweight modules with minimal parameters compared to the primary model.
> - Secondly, training GCN in stage one only requires less than 3 minutes, and the trained GCN can always be reused to guide different primary models on the very same task. Moreover, computing $\mathcal L_{align}$ and $\mathcal L_r$ is fast and will not increase much training time.
>
> The following table shows the comparison between LSG and vanilla fine-tuning (ResNet-50 is adopted as backbone). It is concluded that the extra training cost brought by LSG is not overwhelming.
>
> |Method | Parameters (training) | Parameters (inference) | Training time |
> | -- | :--:|:--:|:--:|
> | Fine-tuning| 25.6M | 25.6M|31.9 min|
> |LSG|28.8M |25.6M|32.2 min|
>
> **Q7. Extension to boarder application range.**
>
> A7. Good comment! In fact, we are actively investigating to extend LSG to image semantic segmentation task. As shown in table below, we conduct preliminary experiments on standard Pascal VOC dataset under SSL setting with different labeling ratio, and find that LSG is also beneficial to data-efficient segmentation. We will leave further studies to future work.
>
> | Network| Method| 1/16 (662 labeled) | 1/8 (1323 labeled) | 1/4 (2645 labeled) |
> | -- | -- | :--: | :--: | :--: |
> | DeepLabV3+ | Suponly                |        64.8        |        68.3        |        70.5        |
> | +          | PseudoLabel (baseline) |        69.4        |        72.1        |        73.9        |
> | ResNet-50  | **LSG (extended)**        |      **71.8**      |      **74.6**      |      **75.3**      |
>
> As for the text task, it may require certain modification of the current method in order to be applied. It is because the current semantic graph is constructed using pre-trained language model and thus may not be much helpful to other language models as the primary model. Still, we point out that alternative ways to build the semantic graph can be proposed and improve model performance on language models using our proposed alignment strategy in section 3.2.

---

> > ### Comment · Reviewer_pDMD · 2023-08-15
> >
> > Thanks for the detailed response! I will raise my confidence score.
> >
> > One remaining question (which also mentioned by some other reviewers) is whether GCN is necessary. Since the training time in stage 1 is so short, the training objective may be quite easy. I am unsure whether GCN uses the embedding as a "shortcut" to do classifications and ignore edges.

---

> > > ### Author Response · Authors · 2023-08-16
> > > **Experiments that proves our GCN does not ignore edges and is necessary**
> > >
> > > Thanks for your valuable feedback. To answer your question, similar to our response to reviewer ZDzt, we would like to demonstrate from three aspects that our GCN classifies nodes **according to both the initial node features and the graph topology** (it does not ignore edges).
> > >
> > > - We conduct a new ablation study as follows.
> > >   - The GCN is firstly trained using standard procedure in our first stage. Then we change the adjacency matrix $A$ of the original graph to an identity matrix $I$ (in other words, we erase all the topology information and leave the node features unchange). We use the trained GCN to predict the nodes on the new graph and denote this variant as *GCN w/ $I$*. **We observe significant accuracy drops compared to the prediction accurcies on the original graph** (*GCN w/ $A$*). This observation shows that the trained GCN actually depends on the adjacency matrix $A$, rather than solely using the initial node features.
> > >   - For a better comparison, we train a linear classifier on the node features, which can be equivalent to a GCN "using the shortcut" as you mentioned. The results in table below show that the linear classifier performs better than our GCN when only node features are provided, yet underperforms the GCN when the topology information is included. Therefore, we conclude that the GCN trained by $\mathcal L_{node}$ does not simply using the shortcut.
> > >   - | Variant|  *Aircraft* | *StanfordCars* | *CUB200*  |
> > > 	| --| :--: | :--: | :--: |
> > > 	| GCN w/ $A$  | 100 | 99.7 | 100 |
> > > 	| GCN  w/ $I$ | 89.7 | 90.9 | 93.5 |
> > > 	|linear probe | 96.4 | 98.8 | 100 |
> > > - We refer back to the results in Fig. 2(d), which are reported here below (the GCN accuracy is tested on a validation graph based on another node feature set). Please note that our designed graph includes a few edges connecting nodes from different classes, where its amount is controlled by cross-label edge ratio. Therefore, different ratio corresponds to different graph topology on top of a same node feature set. We observe that the accuracy of GCN is influenced by the edge ratio, namely the topology information. Moreover, GCN trained from different topology structure has different impacts on the primary model accuracy in the 2nd stage, all proving that GCN pays attention to the edge information.
> > >
> > >   - |Cross-label Edge Ratio|0.1%|0.2%|0.3%|2.0%|
> > >   | -- | :--: | :--: | :--: | :--: |
> > >   | GCN acc. (validation graph) | 99.8 | 99.6 | 99.1 | 75.5 |
> > >   | primary model acc.| 70.7 | 71.2 | 71.3 | 68.7 |
> > >
> > > - The T-SNE visualization also supports our claim. In the original node feature space, there exist a few "prompt clusters" which is hard to discriminate by class. Specifically, each sentence we sent into PLM consists of a prompt template and a label. In some cases, the prompt template dominents in the text embedding and overshaows the class discriminative information. However, the GCN refined embedding space no longer has such prompt cluster, **demonstraing the effect of graph edges that connect these nodes to others sharing same label**.
> > >
> > > Next, we show that the trained GCN is necessary for the 2nd stage via showing full results of the prototype alignment variant.
> > > The prototype alignment refers to aligning image features with prototypes of original label embeddings, therefore it only leverages the embedding information and ignores the graph topology. We see from results that **prototype alignment is less beneficial than our GCN on all datasets**.
> > > |Method| Car-15% | Car-30% | Car-50% | Car-100% | CUB-15% | CUB-30% | CUB-50% | CUB-100% |
> > > |--|--|--|--|--|--|--|--|--|
> > > |Prototype Alignment| 52.4 | 73.0| 82.6| 89.5 | 56.0| 70.1| 76.8| 81.9 |
> > > |LSG| **55.4** | **75.5** | **83.8** | **90.7** | **57.7** | **70.6** | **77.2** | **82.2** |
> > >
> > > | Source domain|Ar|Ar|Ar|Ar|Cl|Cl|Cl|Cl|Pr|Pr|Pr|Pr|Rw|Rw|Rw|Rw| Avg. ID| Avg. OOD|
> > > |--|:--:|:--:| :--: | :--: | :--: | :--: | :---: | :--: | :--: | :--: | :---: | :------: | :------: | :------: | :------: | :------: | :------: | :------: |
> > > | Target domain         | **Ar**|Cl| Pr| Rw  |    **Cl**    |    Ar    |    Pr    |    Rw    |    **Pr**    |    Ar    |    Cl    |    Rw    |**Rw**|Ar|Cl|Pr| Avg. ID| Avg. OOD|
> > > | Prototype Alignment   |   85.0   |   56.9   |   71.3   |   78.1   |   86.2   |   69.5   |   72.9   |   75.9   |   94.3   |   61.9   |   50.4   |   80.6   |90.7|71.8|55.9|81.0|89.0|68.8|
> > > | Contrastive Alignment |   85.6   |   57.1   |   71.6   |   78.5   | **86.3** |   69.9   |   73.0   |   76.7   |   95.0   |   62.5 |   52.1   |   80.7   |90.8|72.2|56.2|81.4|89.4|69.3|
> > > | **LSG**| **85.8** | **57.7** | **74.0** | **79.9** | **86.3** | **71.7** | **75.4** | **77.8** | **95.1** | **65.8** | **54.7** | **82.1** |**91.2**|**73.8**|**58.0**|**82.3**|**89.6**|**71.1**|
> > >
> > > Finally, **$\mathcal L_r$ cannot be applied without GCN in the 2nd stage**, whereas its effectiveness is demonstrated in the ablation study in table 6. In conclusion, we argue that $\mathcal L_{node}$ trained GCN is necessary for our proposed method.

---

> > > > ### Comment · Reviewer_pDMD · 2023-08-19
> > > >
> > > > Thanks for your response.  The additional results make me more confident about the paper. I will keep my score unchanged since I already lean to accept.

---

> > > > > ### Author Response · Authors · 2023-08-20
> > > > > **Thank you**
> > > > >
> > > > > We thank you again for your valuable time and feedback.

---

### Official Review · Reviewer_biPo · 2023-07-06

**Soundness:** 3 good
**Presentation:** 4 excellent
**Contribution:** 3 good
**Rating:** 7
**Confidence:** 5

**Summary:**

The paper introduces the Language Semantic Graph (LSG), a novel approach to data-efficient learning that leverages semantic information from labels. The LSG is used to train an auxiliary graph neural network, which then guides the primary model's training, enhancing the utilization of label knowledge. This method is applicable across various modalities, including image, video, and audio, and has shown significant performance enhancement in both Transfer Learning and Semi-Supervised Learning scenarios.

Experiments were conducted on seven standard datasets covering images, videos, and audios, using several deep neural networks with different architectures and pretraining datasets. The results show that LSG significantly outperforms other methods, especially when labeled data is scarce. It also demonstrates promising potential in semi-supervised settings, achieving the best performance across all labeling rates and datasets. When applied to self-supervised pretrained models, LSG shows consistent gains. It also improves model performance on both in-distribution and out-of-distribution samples, indicating that label semantic relations help the model learn more robust features.

In video and audio experiments, LSG consistently improves the fine-tuning accuracy across all tasks with limited labeled samples. It outperforms other methods, boosting accuracy significantly. For audio experiments, LSG achieves an average of 5.56% accuracy enhancement from the baseline, demonstrating its wide applicability across various modalities.

"LSG consists of two parts: an auxiliary graph neural network that extracts knowledge from the semantic graph and two novel optimization objectives that transfer the knowledge to primary models." The authors demonstrate that LSG is applicable on image, video and audio models and brings significant performance gains to the model under Transfer Learning and Semi-Supervised Learning scenarios.

**Strengths:**

* Innovative Approach: The paper introduces a novel method, the Language Semantic Graph (LSG), which leverages semantic information from labels to guide the training of machine learning models. This approach offers a new perspective on data-efficient learning that has not been extensively explored in previous research.
* Versatility Across Modalities: The LSG method is applicable across various modalities, including image, video, and audio. This wide applicability demonstrates the robustness and flexibility of the proposed method.
* Superior Performance: The LSG method shows significant enhancement in performance compared to other data-efficient learning approaches in both Transfer Learning and Semi-Supervised Learning scenarios. This is a strong indication of the effectiveness of the proposed method.
* Robustness to Data Scarcity: The LSG method performs particularly well when labeled data is scarce, which is a common challenge in machine learning. This makes it a valuable tool for scenarios where data collection and labeling are costly or impractical.
* Improved Out-of-Distribution Performance: The LSG method improves model performance on both in-distribution and out-of-distribution samples. This suggests that the label semantic relations help the model learn features that are more robust to distribution shift, enhancing the model's generalizability.

**Weaknesses:**

The effectiveness of the LSG method relies heavily on the quality and semantic richness of the labels. In scenarios where labels are sparse, ambiguous, or not well-defined, the performance of the LSG method could be compromised.

**Questions:**

*The LSG method relies heavily on the quality and semantic richness of the labels. How does the quality of the labels impact the performance of the LSG method? Could the LSG method be adapted to work effectively with less informative or ambiguous labels, and if so, how?

* The paper primarily focuses on classification tasks. Could the LSG method be adapted or extended to other tasks, and if so, what modifications would be necessary?

**Limitations:**

* The LSG method relies heavily on the quality and semantic richness of the labels. If the labels are not well-defined, sparse, or ambiguous, the performance of the LSG method could be compromised. This dependence on high-quality labels could limit the applicability of the method in certain scenarios.

---

> ### Author Rebuttal · Authors · 2023-08-09
>
> We thank you for your positive feedback and comments on our submission. Please find our responses for your questions below.
>
> **Q1. Performance evaluation on low quality labels.**
>
> A1. Thanks for the good comment. To investigate the effectiveness of LSG on low quality label scenarios, we stimulate the label corruption scenarios by manually adding label noise to the three original datasets in Table 1, following standard protocol in learning from noisy label studies. The results is shown in the following table. (20% noise means that 20% of the training data are randomly assigned to false labels.)
>
> | Variants              | Air-15%  | Air-30%  | Air-50%  | Air-100% | Car-15%  | Car-30%  | Car-50%  | Car-100% | CUB-15%  | CUB-30%  | CUB-50%  | CUB-100% |
> | --------------------- | -------- | -------- | -------- | -------- | -------- | -------- | -------- | -------- | -------- | -------- | -------- | -------- |
> | Fine-tuning  clean    | 41.6     | 57.8     | 68.7     | 80.2     | 41.1     | 65.9     | 78.4     | 87.8     | 51.2     | 64.6     | 74.6     | 81.8     |
> | **LSG clean**         | **55.6** | **72.0** | **79.5** | **86.7** | **55.4** | **75.5** | **83.8** | **90.7** | **57.7** | **70.6** | **77.5** | **82.2** |
> | Fine-tuning 20% noise | 27.2     | 38.8     | 50.7     | 62.6     | 21.7     | 43.6     | 58.1     | 71.8     | 36.2     | 50.1     | 57.4     | 69.7     |
> | **LSG 20% noise**     | **41.5** | **54.8** | **61.9** | **72.6** | **36.3** | **54.3** | **67.9** | **80.8** | **42.1** | **57.3** | **64.2** | **73.8** |
> | Fine-tuning 40% noise | 15.7     | 20.7     | 33.7     | 44.3     | 13.9     | 27.8     | 37.7     | 51.3     | 23.0     | 35.2     | 44.0     | 52.6     |
> | **LSG 40% noise**     | **27.5** | **35.6** | **44.7** | **52.8** | **23.4** | **38.0** | **46.4** | **58.6** | **27.9** | **39.5** | **47.2** | **56.9** |
>
> The results show that both methods are influenced by the label noise and their performance degradation is observed. Still, LSG outperforms fine-tuning baseline significantly on each noise level, and achieves relatively less accuracy degradation from clean label scenarios. The possible reason behind such robustness is that **LSG regularizes the image feature space to maintain the category relationships, which prevents the feature encoder learns heavily distorted feature space by overfitting to the noisy data**. Moreover, methods that identifiy and remove noisy labels can be integrated to LSG on these scenarios to improve its performance.
>
> This experiment shows the potential of using LSG on scenarios with low quality labels. Still, we acknowledge that there are other scenarios where label quality is not satisfying, and we will leave it for future study.
>
> **Q2. Extension to semantic segmentation task.**
>
> A2. Good question! We are actively investigating the extension of LSG to semantic segmentation. Table presented below demonstrates the results of LSG applying on the classical Pascal VOC dataset under semi-supervised learning. The labeling ratio is 1/16, 1/8 and 1/4.  Our current extension involves:
>
> - Both losses $\mathcal L_{align}$ and $\mathcal L_{r}$ are applied on the feature map produced by the ResNet-50 backbone in a similar manner as in classification tasks.
> - Our key modification is to **group the neighboring pixel features together to reduce the number of features added to the language semantic graph each time**, since there will be too many new nodes if every pixel feature is calculated independently for dense prediction tasks. In fact, if all the pixel features are counted individually, the total number of new nodes within the mini-batch will exceed the number of original node of label embeddings, which affects the alignment.
>
> We can observe from the following table that LSG brings improvement to the baseline in data-efficient scenarios for segmentation. As it is just a preliminary study, we will continue the investigation of making extension of the proposed LSG.
>
> |Network| Method                 | 1/16 (662 labeled) | 1/8 (1323 labeled) | 1/4 (2645 labeled) |
> |--| -- | :--: | :--: | :--: |
> |DeepLabV3+| Suponly                |        64.8        |        68.3        |        70.5        |
> |+| PseudoLabel (baseline) |        69.4        |        72.1        |        73.9        |
> |ResNet-50| LSG (extended)         |      **71.8**      |      **74.6**      |      **75.3**      |
>
> We welcome further questions that you find worth discussing!

---

> > ### Comment · Reviewer_biPo · 2023-08-15
> > **Thanks for the update, the authors addressed my concerns well. So I update the score from 6 to 7**
> >
> > Thanks for the update, the authors addressed my concerns well. So I update the score from 6 to 7

---

> > > ### Author Response · Authors · 2023-08-15
> > >
> > > We thank you for your valuable time and feedback.

---

### Official Review · Reviewer_ZDzt · 2023-07-07

**Soundness:** 3 good
**Presentation:** 3 good
**Contribution:** 3 good
**Rating:** 4
**Confidence:** 4

**Summary:**

The typical supervised training ignores the semantic information in the labels. This paper proposes to use the label semantic information during fine-tuning. 1) A label semantic graph is constructed by calculating the sentence embedding similarity of the label descriptions; 2) a GNN is trained on the semantic graph with a node classification objective; 3) the GNN is frozen and guides the training of a visual classifier with two additional loss terms. Specifically, at training time, for each data batch, we can create a graph with the images and labels as nodes; the GNN can be used to encode the graph and the data representations are treated as the initial node features; one loss terms classify the final node features while one loss term minimises the distance between the data representations and the final node features after GNN.

The experiments are extensive, covering image, audio, and video classification. The approach can be used in a semi-supervised setting as well, where the pseudo-label data are only used in the additional loss terms and do not bias the classification head. The performance improvement is significant in many cases.


**Strengths:**

1. The high-level motivation makes sense. It is desirable to utilise the semantic information within the labels during training.

2. The performance improvement is significant in many settings and the evaluation is comprehensive.


**Weaknesses:**

1. While the idea of using label semantic information is good, the presented solution seems ad-hoc and not well motivated.
Conceptually, the new information we introduce is by constructing the semantic graph A with a pre-trained text encoder; after this, it seems rather unclear what motivates the two-step approach with a GNN.
In addition, I find the training objective of the GNN unclear (see Question).

The role of multiple language prompts is also unclear. Is it because using multiple prompts builds a more informative graph?

2. One straightforward way to utilise the label semantic information is to directly use a language encoder to encode the labels and then fine-tune the language encoder along with the data encoder (this is different from the Language Head baseline where the language embeddings are fixed). According to ELEVATAR, this greatly outperforms vanilla fine-tuning.
I wonder if the authors have considered this baseline, especially in the setting of Table 2: can we use CLIP image encoder + CLIP language encoder? Basically, if our pre-trained model already considers the semantic information of the labels (e.g., CLIP), does the approach still bring much improvements?

ELEVATER: A Benchmark and Toolkit for Evaluating Language-Augmented Visual Models


**Questions:**

For training the GNN, isn’t the node classification objective trivial? The objective is basically recovering the concept mentioned in the text prompt, which seems like an easy to solve task; the node feature at the first layer (K^0) should be enough to perfectly solve the task. This is different from the attribute classification task in typical GNN training, where there is not enough information to determine the node label without the adjacency matrix.

**Limitations:**

No evident negative societal impact.

---

> ### Author Rebuttal · Authors · 2023-08-09
>
> We thank you for your endorsement of our idea and the insightful feedback. Please find our responses for your questions below.
>
> **Q1. Motivation of the two-step approach with GNN.**
>
> A1.  We take vision model as an exemplar of the primary encoder $F$  to explain the idea. Our main goal is to align the image features encoded by $F$ towards the semantic space of label embeddings obtained from pre-trained language model. The GNN-based two-step approach promotes the alignment with the following two-fold reason:
>
> - In the first stage, we adopt multiple language prompts to increase the information and diversity of label embeddings for each category. Yet a side effect of is that some poorly-crafted prompt template may generate "prompt cluster" in embedding space due to its dominance influence on the label embedding over the concept that we want to distinguish (see Figure 3(a)). To minimize their negative impact, **GNN with its strength in massage passing, refines the initial node embeddings by leveraging graph topology**. We train GNN to obtain better label embeddings that are both semantic discriminative and structure-preserving. We observe from the T-SNE in Figure 3(a) that the node embeddings after refinement are more discriminative and preserves semantics.
> - More importantly for the second stage, GNN acts as the key component of the proposed alignment method. **By joining the image features as new nodes to the graph, we directly control the knowledge transfer process on the augmented graph via GNN**. To show the superiority of GNN-based alignment method, we conduct an ablation study by considering two classical alignment strategy as alternatives: *Prototype alignment* refers to minimizing the distance between image features and their corresponding category prototype computed from the average of label embeddings; *Contrastive alignment* refers to using supervise contrastive loss that treats label embeddings from the same category as positive samples. The results below validate the superiority of GNN-based alignment.
>
> | Variants| Aircraft-15% | Aircraft-30% | Aircraft-50% | Aircraft-100% |
> | -- | :--: | :--: | :--: | :--: |
> | Prototype alignment | 51.1 | 68.8 | 76.5 | 84.6 |
> | Contrastive alignment | 52.2 | 69.7 |77.3|85.1|
> | LSG | **55.6** | **72.0** | **79.5** | **86.7** |
>
> **Q2. Explanation of the  GNN training objective.**
>
> A2. Thanks for your question. It is a good question, and the answer to it reflects the key technical innovation of LSG. If we only consider the node classification on the original graph $\mathcal G$, leveraging the initial label embeddings is pretty enough. However, the purpose of the GNN is not only about classifying these nodes, but more importantly to guide image features learn label semantics. Therefore, we design a pair of losses $\mathcal L_{node}$ and $\mathcal L_{align}$ that works cooperatively to align image features towards corresponding label embeddings in the graph, which differs from the classical purposes of GNN.
>
> Specifically, due to the large difference between image features and label embeddings at the beginning of the second stage, even a well-trained GNN cannot classify the new nodes correctly. Since the GNN is set fixed, the only way to minimize $\mathcal L_{align}$ is to update $F$ to generate image features more similar to label embeddings, allowing the GNN to recognize them. Therefore, the purpose of loss $\mathcal L_{node}$ is to prepare a GNN ready for $\mathcal L_{align}$ in the second stage. As shown in the following table, if we use a randomly initialized GCN in the second training stage (i.e., w/o $\mathcal L_{node}$), the performance will drop and be similar to the variant where $\mathcal L_{align}$ is excluded.
>
> |Variants | Cars-15% | Cars-30% | Cars-50% | Cars-100% |
> | -- | :--: | :--: | :--: | :--: |
> | LSG w/o $\mathcal L_{node}$ (1st stage) | 48.6 | 68.2 | 80.4 | 89.1 |
> | LSG w/o $\mathcal L_{align}$ (2nd stage) | 48.9 | 68.5 |80.5|89.3|
> | LSG w/ trained GCN| **55.4** | **75.5** | **83.8** | **90.7** |
>
> **Q3. Improved baseline and CLIP$_{text}$ as language model**.
>
> A3. Good comment. As you suggested, we conduct the following experiments under the same setting as table 2 with two new variants.
>
> - *Tunable language head* refers to the stronger baseline by making the language embedding initialized classifier tunable. We draw the same conclusion as in ELEVATER that the tunable head achieves better performance than random initialization. Yet, our method still outperforms this language-augmented baseline on all the tasks with a great margin. We will add this new baseline to the paper.
> - *CLIP$_{text}$* refers to substituting language model from BERT to CLIP language encoder. We find it achieves similar performance to BERT encoder. Reason behind this effectiveness is that, although CLIP image encoder already considers the label knowledge contained in CLIP$_{text}$ in pre-training stage, the fine-tuning process will distort the image feature space [31] and LSG serves as a regularization to prevent the model from forgetting such label semantic knowledge.
>
> Please refer to Table A in the pdf for results.
>
> **Q4. The effect of multiple language prompts.**
>
> A4. We agree with your opinion. Different prompts will put the label concept into different contexts, and consequently produce divergent label embeddings. Such diversity leads to increased information and improved performance. As shown in table below, increasing the number of prompt $m$ leads to better performance. However, consider that the number of nodes in the semantic graph is proportional to $m^2$, and notices that the performance gain is no longer significant when $m>20$, we find a balance between accuracy and training cost by setting $m=20$.
>
> | Task| $m=$5 | 10 | 20 | 40 | 80 |
> | -- | :--: | :--: | :--: | :--: | :--:|
> | Air-15% | 54.8 | 55.0  | 55.6  |55.7| 55.7 |
>
> We hope these responses address your concerns, and welcome any further questions that you find worth discussing!

---

> > ### Comment · Reviewer_ZDzt · 2023-08-15
> > **Method Contribution Still Unclear**
> >
> > Dear authors,
> >
> > Thank you for the detailed rebuttal. For Q2, while I agree that L_align makes sense, my question was mainly targeted at the first-stage training. As you mentioned, when training only with L_node in the first stage, the training objective appears very simple: leveraging the initial label embeddings (K^0) is pretty enough to do node classification; I fail to see why "train GNN to obtain better label embeddings that are both semantic discriminative and structure-preserving". If the node classification can be perfectly performed with only node features at K^0 without information from the adjacency matrix, then I think the network is not encouraged to preserve/utilize the graph structure? I.e., the GCN may simply reduce to a network that acts like it directly predict the class from K^0.
> >
> > Is there some kind of loss/accuracy we could report for the first-stage training? I would image that the model can achieve 100% accuracy easily.
> >
> > In other words, I do not see the benefit of using a GCN and L_align in the second stage (compared to the language head baseline), as I suspect that the GCN may reduce to a very simple node-feature-readout-network since its first-stage objective is easy.

---

> > > ### Author Response · Authors · 2023-08-16
> > > **Experiments that proves our GCN is more than a readout network and is necessary**
> > >
> > > Thanks for your insightful feedback. First of all,  we would like to demonstrate from three aspects that our GCN classifies nodes **according to both the initial node features and the graph topology**.
> > >
> > > - We conduct a new ablation study as follows.
> > >   - The GCN is firstly trained using standard procedure in our first stage. Then we change the adjacency matrix $A$ of the original graph to an identity matrix $I$ (in other words, we erase all the topology information and leave the node features unchange). We use the trained GCN to predict the nodes on the new graph and denote this variant as *GCN w/ $I$*. **We observe significant accuracy drops compared to the prediction accurcies on the original graph** (*GCN w/ $A$*). This observation shows that the trained GCN actually depends on the adjacency matrix $A$, rather than solely using the initial node features.
> > >   - For a better comparison, we train a linear classifier on the node features, which can be viewed as the node-feature-readout-network you mentioned. The results in table below show that the readout network performs better than our GCN when only node features are provided, yet underperforms the GCN when the topology information is included. Therefore, we conclude that the GCN trained by $\mathcal L_{node}$ is more than a readout network.
> > >   - | Variant|  *Aircraft* | *StanfordCars* | *CUB200*  |
> > > 	| --| :--: | :--: | :--: |
> > > 	| GCN w/ $A$  | 100 | 99.7 | 100 |
> > > 	| GCN  w/ $I$ | 89.7 | 90.9 | 93.5 |
> > > 	|linear probe | 96.4 | 98.8 | 100 |
> > > - We refer back to the results in Fig. 2(d), which are reported here below (the GCN accuracy is tested on a validation graph based on another node feature set). Please note that our designed graph includes a few edges connecting nodes from different classes, where its amount is controlled by cross-label edge ratio. Therefore, different ratio corresponds to different graph topology on top of a same node feature set. We observe that the accuracy of GCN is influenced by the edge ratio, namely the topology information. Moreover, GCN trained from different topology structure has different impacts on the primary model accuracy in the 2nd stage, all proving that GCN learns the structure information.
> > >
> > >   - | Cross-label Edge Ratio|0.1%|0.2%|0.3%|2.0%|
> > >   | -- | :--: | :--: | :--: | :--: |
> > >   | GCN acc. (validation graph) | 99.8 | 99.6 | 99.1 | 75.5 |
> > >   | primary model acc.| 70.7 | 71.2 | 71.3 | 68.7 |
> > >
> > > - The T-SNE visualization also supports our claim. In the original node feature space, there exist a few "prompt clusters" which is hard to discriminate by class. Specifically, each sentence we sent into PLM consists of a prompt template and a label. In some cases, the prompt template dominents in the text embedding and overshaows the class discriminative information. However, the GCN refined embedding space no longer has such prompt cluster, **demonstraing the effect of graph edges that connect these nodes to others sharing same label**. Table below compares the Calinski-Harabasz Index between the two embedding space, where higher score means better clustering, and shows GCN learns discriminative info.
> > >   - | Embedding| *OfficeHome* | *Aircraft* | *StanfordCars* | *CUB200*  |
> > >   |--| :--:|:--: | :--: | :--: |
> > >   | Initial Label Embedding |85.5|86.5|137.2|157.8   |
> > >   | GCN refined Embedding | **1395.2**  |**1364.1** |  **1481.7**   | **916.3** |
> > >
> > > Next, we show the loss and accuracy curve of GCN on Aircraft below. GCN finally reaches 100% accuracy on training graph, yet requires a period time of training.
> > >
> > > |Iter |1| 10| 50 |100 | 200 | 500| 1000 | 2000 | 5000 |
> > > |--| --|--| --| --|--|--|--|--|--|
> > > | loss |4.61| 4.55 | 4.08 | 3.95 | 3.86 | 3.78 | 3.72 | 3.66 | 3.63 |
> > > | acc. |0.0| 2.0  | 60.2 | 85.7 | 94.1 | 97.0 | 99.0 | 100  | 100  |
> > >
> > > Finally, we show results of prototype alignment on more datasets. According to your description, we think this prototype alignment is what you mean by "language head baseline", since that the prototypes calculated by averaging node features is a readout network and it **supervises the projected image feature** $h$, replacing our GCN. In contrast, by language head we refer to directly replacing the random initialized classifier by language initialization, therefore is not suitable to ablate GCN. We see from results that a **readout network (prototype alignment) is less beneficial than our GCN**. Moreover, **$\mathcal L_r$ cannot be applied without GCN in the 2nd stage**. Therefore, we argue that $\mathcal L_{node}$ trained GCN is necessary in the 2nd stage. (OH stands for OfficeHome dataset and the detailed results can be found in response to reviewer 3r7Y.)
> > >
> > > |Method| Car-15% | Car-30% | Car-50% | Car-100% | CUB-15% | CUB-30% | CUB-50% | CUB-100% | OH OOD Avg. |
> > > |--|--|--|--|--|--|--|--|--|--|
> > > |Prototype Alignment| 52.4    | 73.0    | 82.6    | 89.5     | 56.0    | 70.1    | 76.8    | 81.9     | 68.9        |
> > > |LSG| **55.4** | **75.5** | **83.8** | **90.7** | **57.7** | **70.6** | **77.2** | **82.2** | **71.1**    |

---

> > > > ### Author Response · Authors · 2023-08-18
> > > > **Looking Forward to Seeing Your New Feedback.**
> > > >
> > > > Dear Reviewer ZDzt,
> > > >
> > > > We are wondering if our newest response resolves your main concern about what the GCN has learned. Specifically, we have provided in our new response strong evidence showing that our GCN indeed requires edge information to make correct predictions. We have also provided comprehensive experiment results on all vision datasets to verify that the trained GCN is more beneficial than a node-feature-readout-network. We are more than delightful to discuss any further questions you have about this key issue. Thanks again for your valuable time and feedback.
> > > >
> > > > Sincerely,
> > > >
> > > > Authors

---

> > > > > ### Comment · Reviewer_ZDzt · 2023-08-20
> > > > >
> > > > > Dear Authors,
> > > > >
> > > > > Thank you for your detailed response and additional results. Indeed, the experiments support that LSG is not "simply" a node-feature-readout network, but I am concerned that it is somehow close to it.
> > > > >
> > > > > The Prototype Alignment achieves quite close performance to LSG (from my observation); thus I am concerned about the benefit of LSG. In addition, the GCN first-stage training converges in 5000 steps and as the authors suggest, "the prompt template dominents in the text embedding and overshadows the class discriminative information"; thus I am worried that the GCN training might be just removing such noise and recovering the information loss, which makes the contribution a bit thin.
> > > > >
> > > > > Thus, I would like to keep my score but I appreciate the rebuttal of the authors.

---

> > > > > > ### Author Response · Authors · 2023-08-21
> > > > > > **Emphasis on two key points**
> > > > > >
> > > > > > We thank you for your valuable time and feedback. We would like to emphasize two points regarding the key issue.
> > > > > >
> > > > > > Stage one is in preparation for the second stage and is designed to be lightweight. The simple yet effective loss $\mathcal L_{node}$ is enough to train a GCN beneficial to the second stage. Please note that our **methodology innovation** of guiding the model learning with semantic graph is largely embodied in the second training stage, which we call ''the primary training stage'' in our paper.
> > > > > >
> > > > > > LSG is significant in performance and widely applicable to tasks of different modalities. To our best knowledge, the 'Prototype Alignment (PA)' variant that appeared in the ablation study during rebuttal has not been explored either, and thus both PA and LSG validate our **conceptual innovation** of learning label semantic knowledge for better representation. And it is shown that LSG is a more sophisticated method than PA.

---

### Author Rebuttal · Authors · 2023-08-10

Dear Reviewers and Area Chair,

We extend our utmost gratitude for your dedicated commitment in meticulously assessing our manuscript and for providing us with your profound insights. Your considered evaluations serve as a testament to the rigor and importance of our research.

We are heartened by the reviewers' recognition of the novelty (biPo, 3r7Y, A9AK), strong motivation (ZDzt, pDMD), clear presentation (biPo, pDMD, A9AK), and significant contribution in the results (ZDzt, biPo, pDMD, 3r7Y, A9AK) of our work.

In our comprehensive individual responses to the reviewers, we have diligently addressed all raised questions and concerns. We eagerly welcome any further inquiries that may arise.

With highest regards,

Authors

---

> ### Comment · Area_Chair_Yazq · 2023-08-21
>
> Thanks for the rebuttal (here and below for each reviewer). We will take it into account during the discussion period.

---

### Author Response · Authors · 2023-08-15
**About the Anonymous Code Link**

Esteemd Area Chair,

It seems that the reviewer cannot get our anonymous code link which is sent previously. I wonder if the code link can be post here where the comment is visible to all the reviewers?

Authors.

---

> ### Comment · Area_Chair_Yazq · 2023-08-15
> **Acknowledgement of Code Link**
>
> Thanks for the message.
>
> -I responded to that reviewer with the code link (that message should be visible to you)
>
> -I also made a general post to the reviewers with the link you sent (that message should not be visible to you).
>
> So you don't need to do anything from your end. Thanks!

---

### Decision · Program_Chairs · 2023-09-21

**Decision:**

Accept (poster)

**Comment:**

The authors propose an interesting idea of using label information via a semantic graph as a regularizer and conduct many experiments across 3 domains (image, video, audio) and 3 out of 5 reviewers support acceptance.

Both reviewers 3r7y and ZDzt expressed the same concern, that it is intuitively unclear why the approach of using a discrete semantic graph based on label embeddings is better than just using the label embeddings directly.

Given the substantial number of experiments provided by the authors in the rebuttal, I favor accepting the paper.